# LEARNING-AUGMENTED MOMENT ESTIMATION ON TIME-DECAY MODELS

**Soham Nagawanshi**[*]      **Shalini Panthangi**[†]      **Chen Wang**[‡]

**David P. Woodruff**[§]      **Samson Zhou**[¶]

## ABSTRACT

Motivated by the prevalence and success of machine learning, a line of recent work has studied learning-augmented algorithms in the streaming model. These results have shown that for natural and practical oracles implemented with machine learning models, we can obtain streaming algorithms with improved space efficiency that are otherwise provably impossible. On the other hand, our understanding is much more limited when items are weighted unequally, for example, in the sliding-window model, where older data must be expunged from the dataset, e.g., by privacy regulation laws. In this paper, we utilize an oracle for the heavy-hitters of datasets to give learning-augmented algorithms for a number of fundamental problems, such as norm/moment estimation, frequency estimation, cascaded norms, and rectangular moment estimation, in the time-decay setting. We complement our theoretical results with a number of empirical evaluations that demonstrate the practical efficiency of our algorithms on real and synthetic datasets.

## 1 INTRODUCTION

The streaming model of computation is one of the most fundamental models in online learning and large-scale learning algorithms. In this model, we consider an underlying frequency vector $\mathbf{x} \in \mathbb{R}^n$, which is initialized to the zero vector $0^n$. The data arrives sequentially as a stream of $m$ updates, where each update at time $t \in [m]$ is denoted by $(t, \sigma_t)$. Each $\sigma_t$ modifies a coordinate $\mathbf{x}_i$ of the frequency vector for some $i \in [n]$ by either increasing or decreasing its value. The goal is usually to compute a function $f(\mathbf{x})$ of this underlying frequency vector using memory substantially smaller than the input dataset size. The data stream model has widespread applications in traffic monitoring (Chen et al., 2021), sensor networks (Gama & Gaber, 2007), data mining (Gaber et al., 2005; Alothali et al., 2019), and video analysis (Xu et al., 2012), to name a few. The research on data streams enjoys a rich history starting with the seminal work of Alon et al. (1999). Some of the most well-studied problems in the data stream model are often related to *frequency (moment) estimation*, i.e., given the frequency vector $\mathbf{x}$, compute the $F_p$ frequency $\|\mathbf{x}\|_p^p = \sum_{i=1}^{n} |\mathbf{x}_i|^p$. A long line of work has thoroughly explored streaming algorithms related to frequency estimation and their limitations (see, e.g., Alon et al. (1999); Chakrabarti et al. (2003); Bar-Yossef et al. (2004); Charikar et al. (2004); Woodruff (2004); Cormode & Muthukrishnan (2005); Indyk & Woodruff (2005); Li (2008); Andoni et al. (2011); Kane et al. (2011); Braverman & Ostrovsky (2013); Braverman et al. (2014; 2018b); Woodruff & Zhou (2021a;b); Indyk et al. (2022); Braverman et al. (2024a) and references therein).

For $p \geqslant 2$, the celebrated count-sketch framework (Charikar et al., 2004; Indyk & Woodruff, 2005) can be used to achieve streaming algorithms that compute a $(1 \pm \varepsilon)$-approximation of the $F_p$ frequency moment in $\widetilde{O}(n^{1-2/p} p^2 / \operatorname{poly}(\varepsilon))$[1] space (Charikar et al., 2004; Indyk & Woodruff, 2005; Andoni et al., 2011). The bound has since been proved tight up to polylogarithmic factors (Chakrabarti

---

[*]Texas A&M University. E-mail: `ndsoham@gmail.com`.

[†]Carnegie Mellon University. E-mail: `shalinipanthangi04@gmail.com`.

[‡]Rensselaer Polytechnic Institute. E-mail: `chen.wang.research@gmail.com`.

[§]Carnegie Mellon University and Google Research. E-mail: `dwoodruf@cs.cmu.edu`.

[¶]Texas A&M University. E-mail: `samsonzhou@gmail.com`.

[1]Throughout, we use $\widetilde{O}(\cdot)$ and $\widetilde{\Omega}(\cdot)$ to hide polylogarithmic terms unless specified otherwise.

et al., 2003; Bar-Yossef et al., 2004; Woodruff, 2004; Woodruff & Zhou, 2021b; Braverman et al., 2024a). As such, for very large $p$, any streaming algorithm would essentially need $\widetilde{\Omega}(n)$ space. The conceptual message is quite pessimistic, and we would naturally wonder whether some beyond-worst-case analysis could be considered to overcome the space lower bound.

**Learning-augmented algorithms.** Learning-augmented algorithms have become a popular framework to circumvent worst-case algorithmic hardness barriers. These algorithms leverage the predictive power of modern machine learning models to obtain some additional "hints". Learning-augmented algorithms have been applied to problems such as frequency estimation (Hsu et al., 2019; Jiang et al., 2020; Chen et al., 2022; Aamand et al., 2025), metric clustering (Ergun et al., 2022; Huang et al., 2025), graph algorithms (Braverman et al., 2024b; Cohen-Addad et al., 2024; Dong et al., 2025; Braverman et al., 2025), and data structure problems (Lin et al., 2022; Fu et al., 2025). Notably, Jiang et al. (2020) showed that for the $F_p$ frequency moment problem with $p \geqslant 2$, with the presence of a natural and practical heavy-hitter oracle, we can obtain $(1 \pm \varepsilon)$-approximation algorithms with $\widetilde{O}(n^{1/2-1/p}/\operatorname{poly}(\varepsilon))$ space – a space bound impossible without the learning-augmented oracle. Jiang et al. (2020) also obtained improved space bounds for related problems such as the rectangle $F_p$ frequency moments and cascaded norms, highlighting the effectiveness of learning-augmented oracles.

**Time-decay streams.** The results in Jiang et al. (2020) and related work (Hsu et al., 2019; Chen et al., 2022; Aamand et al., 2025) gave very promising messages for using learning-augmented oracles in streaming frequency estimation. On the other hand, almost all of these results only focus on estimating the frequencies of the *entire* stream. As such, they do *not* account for the *recency effect* of data streams. In practice, recent updates of the data stream are usually more relevant, and older updates might be considered less important and even invalid. For instance, due to popularity trends, recent songs and movies usually carry more weight on entertainment platforms. Another example of the recency effect is privacy concerns. To protect user privacy, the General Data Protection Regulation (GDPR) of the European Union mandates user data to be deleted after the "necessary" duration (GDPR16). Furthermore, some internet companies, like Apple Inc. (2021), Facebook (2021), Google LLC (2025), and OpenAI (2024) have their own policies on how long user data can be retained.

The time-decay framework in the stream model is a great candidate that captures the recency effect. In this model, apart from the data stream, we are additionally given a function $w$ supported on $[0, 1]$ that maps the importance of the stream updates in the past. In particular, at each time step $t$, we will apply $w$ on a previous time step $t' < t$ to *potentially discount* the contribution of the previous update, i.e., $w(\tau) \leqslant w(1)$ for $\tau \geqslant 1$ [2]. Our goal is to compute a function (e.g., $F_p$) of the frequencies with the weighted stream after the update at the $t$-th time for $t \in [m]$. In general, algorithms for standard data streams do *not* directly imply algorithms in the time-decay model. Therefore, it is an interesting direction to ask *whether the learning-augmented heavy-hitter oracle could be similarly helpful for frequency estimation in time-decay models*.

Typical time-decay models include the *polynomial decay* model (Kopelowitz & Porat, 2005; Cormode et al., 2007; 2009; Braverman et al., 2019), where the importance of the updates decays at a rate of $1/\tau^s$ for some fixed constant $s$, and the *exponential decay* model (Cohen & Strauss, 2003; Cormode et al., 2008; 2009; Braverman et al., 2019), where the decay is much faster as a function of $1/s^\tau$ for some fixed constant $s$. The study of *frequency estimation* often appears in conjunction with the time decay model. In addition to the space bound studied by, e.g., Kopelowitz & Porat (2005); Braverman et al. (2019), several papers have approached the problem from the practical perspective (Xiao et al., 2022; Pulimeno et al., 2021). However, to the best of our knowledge, the general time-decay streaming model has not been well studied in the *learning-augmented* setting.

A notable special case for the time-decay framework is the *sliding-window stream model* (Datar et al., 2002; Lee & Ting, 2006a;b; Braverman & Ostrovsky, 2007; Crouch et al., 2013; Braverman et al., 2015; 2016; 2021; 2018a; 2020; Woodruff & Zhou, 2021a; Borassi et al., 2020; Epasto et al., 2022; Jayaram et al., 2022; Blocki et al., 2023; Woodruff & Yasuda, 2023; Woodruff et al., 2023; Cohen-Addad et al., 2025; Braverman et al., 2026). Here, we are given a window size $W$, and the time-decay function becomes binary: $w(t') = 1$ if $t'$ is within a size-$W$ window (i.e., $t' \geqslant t - W + 1$), and $w(t') = 0$ otherwise. For this special application, Shahout et al. (2024) studied the learning-

---

[2]Here, step $t'$ uses $w(t - t' + 1)$ as the input. In this way, $w$ could be defined as a non-increasing function, i.e., $w(t - t + 1) = w(1)$.

augmented Window Compact Space-Saving (WCSS) algorithm in sliding-window streams. Although a pioneering work with competitive empirical performances, the WCSS algorithm in Shahout et al. (2024) suffers from two issues: $i$). it does *not* give any formal guarantees on the space complexity; and $ii$). for technical reasons, the paper deviates from the heavy-hitter oracle as in Jiang et al. (2020), and instead used a "next occurrence" oracle that is less natural and arguably harder to implement. Specifically, the hard instances in existing lower bounds for streaming algorithms involve identifying $L_p$ heavy-hitters and approximating their contributions to the $F_p$ moment (Woodruff & Zhou, 2021b). Since the "next occurrence" oracle of Shahout et al. (2024) does not perform this task, it is unclear how their approach could be used to improve standard streaming and sketching techniques. Furthermore, it is unclear how their algorithm could be extended to the general time-decay models as we study in the paper. As such, getting results for general time-decay algorithms would imply improved results for the sliding-window model, which renders the open problem more appealing.

**Our results.** In this paper, we answer the open question in the affirmative by devising near-optimal algorithms in the time-decay model (resp. the sliding-window model) for the $F_p$ frequency estimation problem and related problems. Our main results can be summarized as follows (all of the bounds apply to polynomial decay, exponential decay, and the sliding-window settings).

- $F_p$ *frequency:* We give a learning-augmented algorithm that given the heavy-hitter oracle and the frequency vector $\mathbf{x}$ in the stream, computes a $(1 \pm \varepsilon)$-approximation of the $F_p$ frequency $\|\mathbf{x}\|_p^p = \sum_{i=1}^n |\mathbf{x}_i|^p$ in $\widetilde{O}(\frac{n^{1/2-1/p}}{\varepsilon^{4+p}} \cdot p^{1+p})$ space.

- *Rectangle $F_p$ frequency:* When the universe is $[\Delta]^n$ and stream elements update all coordinates in hyperrectangles, the $F_p$ frequency moment problem for the stream is called *rectangle $F_p$ frequency.* We give a learning-augmented algorithm that computes a $(1 \pm \varepsilon)$-approximation of the rectangle $F_p$ frequency in $\widetilde{O}(\frac{\Delta^{d(1/2-1/p)}}{\varepsilon^{4+p}} \cdot \text{poly}(\frac{p^p}{\varepsilon}, d))$ space with heavy-hitter oracles.

- $(k, p)$-*cascaded norm:* As a generalization of the $F_p$ frequency moment problem, when the frequency data is given as an $n \times d$ matrix $\mathbf{X}$ and the stream updates each coordinate $\mathbf{X}_{i,j}$, we define $f(\mathbf{X}) = (\sum_{i=1}^n (\sum_{j=1}^d \|\mathbf{X}_{i,j}\|^p)^{k/p})^{1/k}$ as the $(k, p)$-cascaded norm ($k$-norm of the $p$-norms of the rows). We give a learning-augmented algorithm that computes a $(1 \pm \varepsilon)$-approximation of the $(k, p)$-cascaded norm in space $\widetilde{O}_{k,p}(n^{1-\frac{1}{k}-\frac{p}{2k}} \cdot d^{\frac{1}{2}-\frac{1}{p}})$. We use $\widetilde{O}_{k,p}(\cdot)$ to hide polynomial terms of $(kp)^{kp}$, $\varepsilon$, and $\log n$. [3]

By a lower bound in Jiang et al. (2020), any learning-augmented streaming algorithm that obtains a $(1 \pm \varepsilon)$-approximation for the $F_p$ moment would require $\Omega(n^{1/2-1/p}/\varepsilon^{2/p})$ space. Since the streaming setting can be viewed as a special case for the time-decay model, our algorithm for $F_p$ frequency is optimal with respect to the exponent $n$.

**Our techniques.** Our approach is fundamentally different from previous work in learning-augmented sliding-window algorithms, e.g., Shahout et al. (2024). In particular, we considered an approach that directly transforms streaming algorithms into time-decay algorithms. Crucially, we observe that many approaches in the time-decay streaming literature are based on *smoothness* of functions (e.g., Braverman & Ostrovsky (2007); Braverman et al. (2019)). Roughly speaking, these approaches follow a framework to maintain *multiple copies* of the streaming algorithm on different *suffixes*, and delete the copies that are considered "outdated". The correctness of time-decay streams could follow if the function satisfies some "smoothness" properties. We derived several *white-box* adaptations of the algorithms under this framework. We show that as long as the learning-augmented oracle is suffix-compatible, i.e., it is able to predict the heavy hitters of suffix streams $[t : m]$ as well, the framework would work in the learning-augmented setting in the same way as the setting without the oracle. As such, we could apply the streaming learning-augmented algorithm in Jiang et al. (2020) to obtain the desired time-decay algorithms.

We remark that another valid option would be to generalize the difference estimator framework of Woodruff & Zhou (2021a) to incorporate advice. Although this approach gives better dependencies in $\varepsilon$, the overall algorithm is quite involved and not as easily amenable to implementation, which in some sense is the entire reason to incorporate machine learning advice in the first place. We thus focus on practical implementations with provable theoretical guarantees.

---

[3]For the polynomial and exponential-decay models, the computation of the $(k, p)$-cascaded norm requires row arrival. For the sliding-window streams, the updates can be on the points.

| Task | Space Bound | Model | Remark |
|------|-------------|-------|--------|
| $F_p$ Frequency ($p \geqslant 2$) | $\tilde{O}(n^{1/2-1/p}/\varepsilon^4)$ | Streaming | Jiang et al. (2020) |
| | $\Omega(n^{1/2-1/p}/\varepsilon^{2/p})$ | Any | Lower bound, Jiang et al. (2020) |
| | not specified | Sliding Window | Shahout et al. (2024) |
| | $\tilde{O}(n^{1/2-1/p} \cdot p^{1+p}/\varepsilon^{4+p})$ | General Time-decay (e.g., Sliding Window) | This work, Theorem 1 |
| Rectangle $F_p$ Frequency | $\tilde{O}(\Delta^{d(1/2-1/p)} \cdot \mathrm{poly}(\frac{d}{\varepsilon}, d))$ | Streaming | Jiang et al. (2020) |
| | $\tilde{O}(\Delta^{d(1/2-1/p)} \cdot \mathrm{poly}(\frac{p^p}{\varepsilon^p}, d))$ | General Time-decay (e.g., Sliding Window) | This work, Theorem 5 |
| $(k, p)$-Cascaded Norm | $\tilde{O}(n^{1-\frac{1}{k}-\frac{p}{2k}} \cdot d^{\frac{1}{2}-\frac{1}{p}})$ | Streaming | Jiang et al. (2020) |
| | $\tilde{O}_{k,p}(n^{1-\frac{1}{k}-\frac{p}{2k}} \cdot d^{\frac{1}{2}-\frac{1}{p}})$ | General Time-decay (e.g., Sliding Window) | This work, Theorem 6 |

Table 1: Summary of the results and their comparisons with existing work. The theorem pointers are directed to the *sliding-window* algorithms as an illustration.

**Experiments.** For the special case of sliding-window streams, we conduct experiments for learning-augmented $F_p$ frequency estimation. We implement multiple suffix-compatible heavy-hitter oracles, such as the count-sketch algorithm (Charikar et al., 2004): this allows us to compute heavy hitters for different suffixes of streams with minimal space overhead. We tested the $F_p$ frequency algorithms based on Algorithm 2 and the implementations in Alon et al. (1999) (AMS algorithm) and Indyk & Woodruff (2005) with and without the learning-augmented oracles. The datasets we tested on include a synthetic dataset sampling from binomial distributions and the real-world internet datasets of CAIDA and AOL. Our experiments show that the learning-augmented approach can significantly boost the performance of the frequency estimation algorithms, and, at times, produce results extremely close to the ground-truth. Furthermore, our approach is fairly robust against distribution shifts over updates, while other heuristic approaches like scaling would induce performance degradation when the distribution changes. [4]

## 2 PRELIMINARIES

**The time-decay and sliding-window models.** We specify the time-decay model and related notation. We assume the underlying data $\mathbf{x} \in \mathbb{Z}^n$ to be a *frequency vector*, where each coordinate $\mathbf{x}_i$ stands for the frequency of the corresponding item. The frequency vector is initialized as $0^n$, and at each time, the vector is updated as $(i, \Delta)$ such that $\mathbf{x}_i \leftarrow \mathbf{x}_i + \Delta$, for some $\Delta \geqslant 0$, so that all updates can only increase the coordinates of the frequency vector. We assume in this paper without loss of generality that $\Delta = 1$. Let $m$ be the total number of updates in the stream, which we assume to be upper bounded by at most some polynomial in the universe size $n$. In the time-decay model, we are additionally given a weight function $w : \mathbb{R} \to \mathbb{R}^{\geqslant 0}$, and an update at time $t \in [m]$ contributes weight $w(m-t+1)$ to the weight of coordinate $i_t \in [n]$. Here, $w$ is a non-increasing function with $w(1) = 1$, and $s > 0$ is some parameter that is fixed before the data stream begins. We have $w(\tau) = 1/\tau^s$ for polynomial decay and $w(\tau) = s^\tau$ for exponential decay, respectively. The underlying weighted frequency vector of the $t$-th time for all $i \in [n]$ is defined as $\mathbf{x}_i^t = \sum_{t' \in [t]: i_{t'} = i} w(t - t' + 1)$.

In the special case of the sliding-window model, we are additionally given a window size $W$. At each step $t \in (W - 1, m]$, we define $\mathbf{x}^{W,t}$ as the frequency vector over the *last $W$ update steps*. Let $f : \mathbb{R}^n \to \mathbb{R}$ be a given function, and our goal is to output $f(\mathbf{x}^{W,t})$ for all $t \in (W - 1, m]$.

---

[4]The codes for the experiments are available at https://github.com/ndsoham/learning-augmented-fp-time-decay

Apart from the subvectors $\mathbf{x}^{W,t}$ which we aim to compute, we also define $\mathbf{x}^{t_1:t_2}$ as the vector obtained by accounting for all the updates from step $t_1$ to $t_2$. In particular, $\mathbf{x}^{1:t_1}$ and $\mathbf{x}^{t_1:m}$ represent the frequency vectors with the updates from the start of the stream to $t_1$ and from $t_1$ till the end of the stream, respectively.

**The functions to compute.** We aim to compute the following objective functions (defined as mappings $\mathbb{R}^n \to \mathbb{R}$) in the sliding-window streaming model.

- The $F_p$ *frequency* function: $f(\mathbf{x}) = \|\mathbf{x}\|_p^p = \sum_i |\mathbf{x}_i|^p$.
- The *rectangle $F_p$ frequency* function: this is a special case for the $F_p$ frequency problem, where we assume $\mathbf{x} \in [\Delta]^n$ for some integer $\Delta$.

Furthermore, we also study the *cascaded norm* function for *high-dimensional frequencies*, i.e., the input "frequency" is a $n \times d$ matrix $\mathbf{X}$, where each row corresponds to a generalized notion of frequency. The $(k, p)$-*cascaded norm* function $f : \mathbb{R}^{n \times d} \to \mathbb{R}$ is defined as $f(\mathbf{X}) = \left( \sum_{i=1}^n \left( \sum_{j=1}^d |\mathbf{X}_{i,j}|^p \right)^{k/p} \right)^{1/k}$. In the streaming model, at each time $t$, an update on a coordinate $\mathbf{X}_{i,j}$ is given in the stream. We can define the corresponding inputs for time-decay and sliding-window models analogously.

**The learning-augmented framework.** We work with learning-augmented streaming algorithms, where we assume an oracle that could predict whether $\mathbf{x}_i$ is a *heavy hitter*. Depending on the function $f$, we have multiple ways of defining heavy hitters as follows.

**Definition 1** (Heavy-hitter oracles). We say an element $\mathbf{x}_i$ is a heavy hitter with the following rules.

(a). If $f$ is $F_p$ *frequency*, we say that $\mathbf{x}_i$ is a heavy hitter if $|\mathbf{x}_i|^p \geqslant \frac{1}{\sqrt{n}} \cdot \|\mathbf{x}\|_p^p$.

(b). If $f$ is *rectangle $F_p$ frequency* in $[\Delta]^d$, we say that $\mathbf{x}_i$ is a heavy hitter if $|\mathbf{x}_i|^p \geqslant \|\mathbf{x}\|_p^p / \Delta^{d/2}$.

(c). If $f$ is $(k, p)$-*cascaded norm*, we say that $\mathbf{X}_{i,j}$ is a heavy hitter if $|\mathbf{X}_{i,j}|^p \geqslant \|\mathbf{X}\|_p^p / (d^{1/2} \cdot n^{1-p/2k})$, where $\|\mathbf{X}\|_p^p$ is the vector norm of the vector from the elements in $\mathbf{X}$.

A heavy hitter oracle $\mathcal{O}$ is a learning-augmented oracle that, upon querying $\mathbf{x}_i$, answers whether $\mathbf{x}_i$ satisfies the heavy hitter definition. We say that $\mathcal{O}$ is a *deterministic* oracle if it always correctly predicts whether $\mathbf{x}_i$ is a heavy hitter. In contrast, we say that $\mathcal{O}$ is a *stochastic* oracle with success probability $1-\delta$ if for each coordinate $\mathbf{x}_i$, the oracle returns whether $\mathbf{x}_i$ is a heavy hitter independently with probability at least $1 - \delta$.

For the purpose of time-decay algorithms, we also need the oracle to be *suffix compatible*, i.e., able to return whether $\mathbf{x}_i$ is a heavy hitter for all suffix streams $[t : m], t \in (0, m - 1]$. We formally define such oracles as follows.

**Definition 2** (Suffix-compatible heavy-hitter oracles). We say a heavy hitter oracle $\mathcal{O}$ is a deterministic (resp. randomized) suffix-compatible learning-augmented oracle if for each suffix of stream $[t : m]$ for $t \in (0, m - 1]$ and each frequency vector $\mathbf{x}(t : m)$, $\mathcal{O}$ is able to answer whether $\mathbf{x}(t : m)_i$ is a heavy hitter (resp. with probability at least $1 - \delta$).

**Additional discussions about suffix-compatible heavy-hitter oracles.** Learning-augmented algorithms with heavy-hitter oracles were explored by Jiang et al. (2020). Our setting is consistent with theirs, and similar to Jiang et al. (2020), such oracles are easy to implement for practical purposes. In Appendix D, we provide a general framework for the learning of such oracles.

We note that Shahout et al. (2024) discussed certain difficulties for using bloom filters to obtain predictions for *every window*. We emphasize that the suffix-compatibility property does *not* require the prediction for every window, but rather only the suffixes of the streams (only $m - W + 1$ such windows). This is a much more relaxed setting than the issues discussed in Shahout et al. (2024). Furthermore, our experiments in Section 5 show that the suffix-compatible oracles can be easily learned via a small part of the streaming updates.

## 3 ALGORITHM AND ANALYSIS FOR THE SLIDING-WINDOW MODEL

We first discuss the special case of *sliding-window* algorithms since the algorithm and analysis are clean and easy to present. For this setting, we take advantage of the *smooth histogram* framework

introduced by Braverman & Ostrovsky (2007). At a high level, a function $f$ is said to be smooth if the following condition holds. Let $\mathbf{x}_A$ and $\mathbf{x}_B$ be two frequency vectors for elements in data streams $A$ and $B$, where $B$ is a suffix of $A$. If $f(\mathbf{x}_A)$ and $f(\mathbf{x}_B)$ are already sufficiently close, then they remain close under any common suffix of updates, i.e., by appending $C$ to both $A \cup C$ and $B \cup C$ and getting new frequency vectors $\mathbf{x}_{A \cup C}$ and $\mathbf{x}_{B \cup C}$, the difference between $f(\mathbf{x}_{A \cup C})$ and $f(\mathbf{x}_{B \cup C})$ remains small. Braverman & Ostrovsky (2007) already established the smoothness of $F_p$ frequencies, and we further prove the smoothness properties for rectangle $F_p$ frequencies and cascaded norms.

We now discuss the framework in more detail, starting with the introduction of the notion of *common suffix-augmented frequency vectors*.

**Definition 3** (Common suffix-augmented frequency vectors). Let $\mathbf{x}_A$ and $\mathbf{x}_B$ be frequency vectors obtained from a stream $A$ with suffix $B$. Furthermore, let $\mathbf{x}_C$ be the frequency vector of a common suffix $C$ of $A$ and $B$. We say that $\mathbf{x}_{A \cup C}$ and $\mathbf{x}_{B \cup C}$ are a pair of common suffix-augmented frequency vectors if $\mathbf{x}_{A \cup C} = \mathbf{x}_A + \mathbf{x}_C$ and $\mathbf{x}_{B \cup C} = \mathbf{x}_B + \mathbf{x}_C$.

In other words, let $A, B, C$ be the streaming elements of $\mathbf{x}_A$, $\mathbf{x}_B$, and $\mathbf{x}_C$, respectively. In Definition 3, we have $A \subseteq B$, and the streaming elements of $\mathbf{x}_{A \cup C}$ and $\mathbf{x}_{B \cup C}$ are $A \cup C$ and $B \cup C$. We are now ready to formally define $(\alpha, \beta)$-smooth functions as follows.

**Definition 4** ($(\alpha, \beta)$-smooth functions, Definition 1 of Braverman & Ostrovsky (2007)). A function $f : \mathbb{R}^n \to \mathbb{R}$ for frequency vectors is $(\alpha, \beta)$-smooth if the following properties hold.

- $f(\mathbf{x}) \geqslant 0$ for any frequency vector $\mathbf{x}$.
- $f(\mathbf{x}) \leqslant \text{poly}(n)$ for some fixed polynomial.
- Let $\mathbf{x}_B$ be a frequency vector obtained from a suffix of $\mathbf{x}_A$, we have
  (1) $f(\mathbf{x}_A) \geqslant f(\mathbf{x}_B)$.
  (2) For any $\varepsilon \in (0, 1)$, there exits $\alpha = \alpha(f, \varepsilon)$ and $\beta = \beta(f, \varepsilon)$ such that
     ○ $0 \leqslant \beta \leqslant \alpha < 1$.
     ○ If $f(\mathbf{x}_B) \geqslant (1 - \beta)f(\mathbf{x}_A)$, then for any common suffix-augmented frequency vectors $\mathbf{x}_{A \cup C}$ and $\mathbf{x}_{B \cup C}$ as prescribed in Definition 3, there is $f(\mathbf{x}_{B \cup C}) \geqslant (1 - \alpha)f(\mathbf{x}_{A \cup C})$.

Using the definition of $(\alpha, \beta)$-smooth functions, we are able to derive the following framework that transform streaming algorithms to sliding-window algorithms in the learning-augmented regime.

**Lemma 3.1.** *Let $f$ be an $(\alpha, \beta)$-smooth function, and let* ALG *be any learning-augmented streaming algorithm that queries the heavy-hitter oracle $\mathcal{O}$ with the following properties:*

- ALG *outputs $f'(\mathbf{x})$ such that $(1 - \varepsilon) \cdot f(\mathbf{x}) \leqslant f'(\mathbf{x}) \leqslant (1 + \varepsilon) \cdot f(\mathbf{x})$ by the end of the stream with probability at least $1 - \delta$;*

- *$\mathcal{O}$ satisfies the suffix-compatible property as prescribed by Definition 2; and*

- ALG *uses $g(\varepsilon, \delta)$ space and performs $h(\varepsilon, \delta)$ operations per stream update.*

*Then, there exists a sliding-window streaming algorithm* ALG$'$ *that computes a $(1 \pm (\alpha + \varepsilon))$-approximation of the sliding-windows with probability at least $1 - \delta$ using $O\left(\frac{(g(\varepsilon, \delta') + \log n) \cdot \log n}{\beta}\right)$ space and $O\left(\frac{h(\varepsilon, \delta') \log n}{\beta}\right)$ operations per stream update, where $\delta' = \frac{\delta \beta}{\log n}$.*

We defer the discussion and the proof for Lemma 3.1 to Appendix A. Next, we show how to use the framework in Lemma 3.1 to obtain learning-augmented sliding-window algorithms. We remark that the bound provided by the framework is *independent of* the window size $W$. As such, our bounds in this section do *not* include the $W$ parameter. Limited by space, we only present our results for $F_p$ frequency moment estimation in this section, and defer the results for rectangle $F_p$ frequency moment estimation and cascaded norms to Appendix A.

Our learning-augmented sliding-window algorithm for $F_p$ estimation for $p > 2$ has the following guarantees with both *perfect* and *erroneous* oracles.

**Theorem 1** (Learning-augmented $F_p$ frequency moment algorithm). *There exists a sliding-window streaming algorithm that, given a stream of elements in a sliding window, a fixed parameter $p \geqslant 2$, and a deterministic suffix-compatible heavy-hitter oracle $\mathcal{O}$ (as prescribed by Definition 2), with probability at least $99/100$ outputs a $(1 + \varepsilon)$-approximation of the $F_p$ frequency in $O\left(\frac{n^{1/2 - 1/p}}{\varepsilon^{4 + p}} \cdot p^{1 + p} \cdot \log^4 n \cdot \log(\frac{p}{\varepsilon})\right)$ space.*

**Theorem 2** (Learning-augmented $F_p$ frequency algorithm with stochastic oracles). *There exists a sliding-window streaming algorithm that, given a stream of elements in a sliding window, a fixed parameter $p \geqslant 2$, and a stochastic suffix-compatible heavy-hitter oracle $\mathcal{O}$ with success probability $1 - \delta$ (as prescribed by Definition 2), with probability at least $99/100$ outputs a $(1+\varepsilon)$-approximation of the $F_p$ frequency moment in space*

- $O\left( \frac{(n\delta)^{1-1/p}}{\varepsilon^{4+p}} \cdot p^{1+p} \cdot \log^4 n \cdot \log(\frac{p}{\varepsilon}) \right)$ *bits if* $\delta = \Omega(1/\sqrt{n})$.
- $O\left( \frac{n^{1/2-1/p}}{\varepsilon^{4+p}} \cdot p^{1+p} \cdot \log^4 n \cdot \log(\frac{p}{\varepsilon}) \right)$ *bits if* $\delta = O(1/\sqrt{n})$.

For any constant $p$ and $\varepsilon$, the space bound for the sliding-window $F_p$ frequency moment estimation problem becomes $\widetilde{O}(n^{1/2-1/p})$ for any learning-augmented oracle with a sufficiently high success probability. This bound is optimal even in the streaming setting by Jiang et al. (2020); as such, we obtain a near-optimal algorithm for the learning-augmented $F_p$ frequency estimation problem.

At a high level, the algorithm is an application of the learning-augmented $F_p$ frequency streaming algorithm in Jiang et al. (2020) to the framework we discussed in Lemma 3.1. The guarantees of the algorithm in Jiang et al. (2020) can be described as follows.

**Lemma 3.2** (Jiang et al. (2020)). *For any given stream, a fixed parameter $p \geqslant 2$, and a stochastic heavy-hitter oracle $\mathcal{O}$ with success probability $1 - \delta$, with probability at least $99/100$, Algorithm 2 computes a $(1 + \varepsilon)$-approximation of the $F_p$ frequency using space*

- $O\left( \frac{(n\delta)^{1-1/p}}{\varepsilon^4} \cdot \log^2 n \right)$ *bits if* $\delta = \Omega(1/\sqrt{n})$.
- $O\left( \frac{n^{1/2-1/p}}{\varepsilon^4} \cdot \log^2 n \right)$ *bits if* $\delta = O(1/\sqrt{n})$.

To apply the reductions of Proposition 2 (presented in the Appendix A), we need to understand the smoothness of the $F_p$ frequency function, which is a standard fact established by previous results.

**Lemma 3.3** (Braverman & Ostrovsky (2007)). *The $F_p$ frequency function is $(\varepsilon, \varepsilon^p/p^p)$-smooth.*

***Finalizing the proof of Theorems 1 and 2.*** We apply Lemma 3.1 to the algorithm of Lemma 3.2 with the smoothness guarantees as in Lemma 3.3. For the success probability, we argue that the algorithm in Lemma 3.2 could be made to succeed with probability at least $1 - \delta$ with $O(\log(1/\delta))$ multiplicative space overhead. This can be accomplished by the classical median trick: we run $O(\log(1/\delta))$ copies of the streaming algorithm, and take the median of the frequency output. By a Chernoff bound argument, the failure probability is at most $\delta$.

Let $\beta = \varepsilon^p/p^p$; for the deterministic oracle case, we could use $g = \frac{n^{1/2-1/p}}{\varepsilon^4} \cdot \log^2 n$ and failure probability $\delta' = O(\beta/\log n)$ to obtain that the space needed for the streaming algorithm is at most

$$g(\varepsilon, \delta') = O\left( \frac{n^{1/2-1/p}}{\varepsilon^4} \cdot \log^2 n \cdot \log(n/\beta) \right) \leqslant O\left( \frac{n^{1/2-1/p}}{\varepsilon^4} \cdot p \cdot \log^3 n \cdot \log(p/\varepsilon) \right).$$

Therefore, the space we need is $O\left( g(\varepsilon, \delta') \cdot \frac{\log n}{\beta} \right) = O\left( \frac{n^{1/2-1/p}}{\varepsilon^{4+p}} \cdot p^{1+p} \cdot \log^4 n \cdot \log(\frac{p}{\varepsilon}) \right)$, as desired. Finally, for the case with the randomized learning-augmented oracle, we simply replace the $n^{1/2-1/p}$ term in the bound with $(n\delta)^{1-1/p}$, which would give us the desired statement. □

## 4    THE ALGORITHMS AND ANALYSIS FOR GENERAL TIME DECAY MODELS

In this section, we describe our algorithms for the general time-decay setting, where an update at time $t' \in [m]$ contributes weight $w(t - t' + 1)$ to the weight of coordinate $i_t \in [n]$ at time $t$. The underlying weighted frequency vector $\mathbf{x}^t$ at step $t$ is defined as

$$\mathbf{x}_i^t = \sum_{t' \in [t]: i_t = i} w(t - t' + 1).$$

The formal definition of the time-decay model and functions could be found in Section 2. Given a non-decreasing function $G : \mathbb{R} \to \mathbb{R}^{\geqslant 0}$ with $G(0) = 0$, we define $G$-moment estimation as the

problem of estimating

$$G(\mathbf{x}) = \sum_{i \in [n]} G(\mathbf{x}_i).$$

When the context is clear, we also use the lower-case $x$ as the input to the function $G$.

We introduce a general framework that transforms a linear sketch streaming algorithm for the problem of $G$-moment estimation into a time decay algorithm for $G$-moment estimation.

**Definition 5** (Smoothness in time-decay models). Given $\varepsilon > 0$, the weight function $w$ and the $G$-moment function $G$ are said to be $(\varepsilon, \nu, \eta)$-smooth if:

(1) $G((1 + \eta)x) - G(x) \leqslant \frac{\varepsilon}{4} \cdot G(x)$ for all $x \geqslant 1$.

(2) There exists an integer $m_\nu \geqslant 0$ such that $\sum_{i \in [m_\nu, m]} w(i) \leqslant \nu$ and $G(x + \nu) - G(x) \leqslant \frac{\varepsilon}{4} \cdot G(1)$ for all $x \geqslant 1$. In other words, all updates in a stream of length $m$ that arrived more than $m_\nu$ previous timesteps can be ignored.

The next theorem provides a framework for polynomial-decay and exponential-decay models; we defer its proof to Appendix B.

**Theorem 3.** *Given a streaming algorithm that provides a $(1 + \varepsilon)$-approximation to $G$-moment estimation using a linear sketch with $k$ rows, functions $G$ and $w$ that satisfy the $(\varepsilon, \nu, \eta)$-smoothness condition (Definition 5), there exists an algorithm for general time-decay that provides a $(1 + \varepsilon)$-approximation to $G$-moment estimation that uses at most $O\left(\frac{k}{\eta} \log n \log \frac{1}{\nu}\right)$ bits of space.*

*Furthermore, the statement holds true for learning-augmented algorithms as long as the oracle $\mathcal{O}$ is suffix-compatible.*

We can apply Theorem 3 on the $F_p$ moment estimation for the polynomial-decay model. We have described in Section 3 the algorithm for $F_p$ estimation (Proposition 3), and we could apply Theorem 3 to Proposition 3 to obtain the following result.

**Theorem 4.** *Given a constant $p > 2$, an accuracy parameter $\varepsilon \in (0, 1)$, and a heavy-hitter oracle $\mathcal{O}$ for the data stream, there exists a one-pass algorithm that outputs a $(1 + \varepsilon)$-approximation to the rectangular $F_p$ moment in the polynomial-decay model that uses $\widetilde{O}\left(\frac{\Delta^{d(1/2 - 1/p)}}{\varepsilon^{2 + 4/p}}\right)$ bits of space.*

## 5 EMPIRICAL EVALUATIONS

**Experimental setup.** To demonstrate the practicality of our theoretical results, we compared non-augmented sliding window algorithms with their augmented counterparts. We implemented Alon et al. (1999)'s algorithm for $\ell_2$ norm estimation, which we refer to as AMS, and Indyk & Woodruff (2005)'s subsampling (SS) algorithm for $\ell_3$ norm estimation. We utilized three different oracles for our augmented algorithms: Charikar et al. (2004)'s CountSketch (CS) algorithm, a ChatGPT/Google Gemini Large Language Model (LLM), and an LSTM trained for heavy hitter prediction. We refer to the augmented algorithms as AMSA and SSA, respectively.

**Datasets.** We tested our implementations on three datasets:

(1) A synthetic, skewed random-integer distribution generated by sampling a binomial distribution with $p = 2 \cdot q/\sqrt{n}$, where $q$ is the desired number of heavy hitters and $n$ is the number of distinct integer values, i.e. the universe size.

(2) CAIDA dataset[5], which contains 12 minutes of IP traffic; each minute contains about 30M IP addresses. We used subsets of the first minute to test our implementations. Each IP address was converted to its integer value using Python's ipaddress library.

(3) AOL dataset [6], which contains 20M user queries collected from 650k users. Each query was associated with an anonymous user id; we used a subset of these ids to test our implementations.

---

[5] https://www.caida.org/catalog/datasets/passive_dataset
[6] https://www.kaggle.com/datasets/dineshydv/aol-user-session-collection-500k

Limited by space, we only show the results for the real-world CAIDA and AOL datasets, and defer the results for the synthetic datasets and provide additional experiments details in Appendix C.

**Computing devices.** We implemented our experiments in Python 3.10.18. The experiments were performed on an Apple MacBook Air M2 with 16GB of RAM. Experiments on CAIDA that varied window sizes took 1100 - 1400 minutes. Experiments that varied sample selection probabilities took 400 - 600 minutes. The runtimes were similar for the AOL dataset.

## 5.1 AMS AND LEARNING-AUGMENTED AMS ALGORITHMS

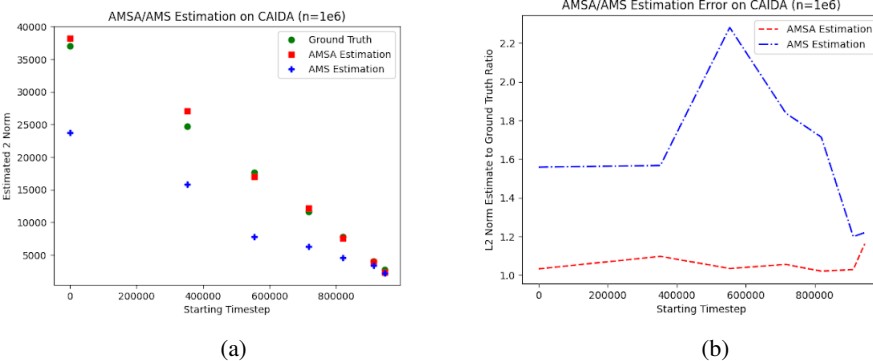

(a)                                                    (b)

Fig. 1: Experiments for $\ell_2$ norm estimation on CAIDA. *Note on the notation:* the variable $n$ in figures refers to stream length (which is $m$ at other places of the paper) .

Figure 1a compares the estimation results from our two algorithms to the actual $\ell_2$ norm over various timesteps. Note that $n$ refers to stream length (not universe size) in Figure 1a, Figure 3a, and Figure 5a. The "timesteps" on the $x$ axis correspond to window sizes: estimates starting at timestep 0 consider all 1M stream items, i.e. $W = m$, while subsequent estimates use the most recent $1M -$ timestep values. In other words, $W = m - t_i$, where $t_i$ is a timestep and $m$ is stream length. As seen in Figure 1a, AMSA estimates are much closer to the ground truth than AMS estimates over all selected window sizes, indicating that the augmented algorithm consistently produces a more accurate estimate. This is further validated by Figure 1b, which plots the ratio of estimates to the ground truth over various window sizes. As shown, AMSA estimates are within a factor of 1.2 over all window sizes, while AMS estimates vary between factors of about 1.25 and 2.3. Additionally, AMSA estimates deviate from the ground truth by a somewhat consistent margin, while AMS estimations seem to get closer to the ground truth as the window size decreases, indicating that the augmented algorithm is more precise over a variety of window sizes. AMSA's precision is confirmed by the flatness of its error curve in Figure 1b over all window sizes, especially when compared to the AMS error line.

## 5.2 SS AND LEARNING-AUGMENTED SS ALGORITHMS

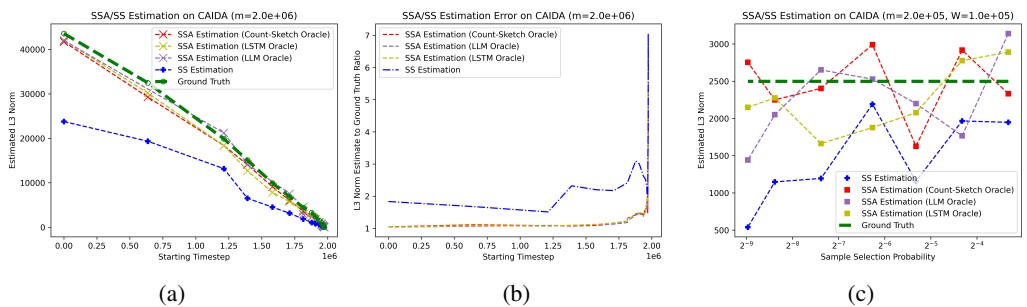

(a)                                (b)                                (c)

Fig. 2: Experiments for $\ell_3$ estimation on CAIDA using multiple oracles

Figure 2a compares the estimation results from SSA and SS to the actual $\ell_3$ norm over various window sizes. All three oracles provide useful augmentation, allowing SSA to estimate norms much

closer to the ground truth than the baseline SS algorithm over all window sizes. Like for AMSA, the SSA error curve is closer to 1 and flatter than the SS error curve, indicating that SSA is more accurate and precise than its non-augmented counterpart.

Figure 2c compares the estimation results from SSA and SS to the actual $\ell_3$ norm over various sample selection probabilities. Again, all three oracles help augment the baseline algorithm, providing an estimation closer to the ground truth across all selection probabilities. Using a higher selection probability roughly corresponds to higher memory usage since more elements must be stored in the estimate sample. The SSA estimation performs particularly well for very low memory usage and largely outperforms SS for higher memory usage, meaning that augmentation is beneficial even when the baseline algorithm has larger estimate sample sizes.

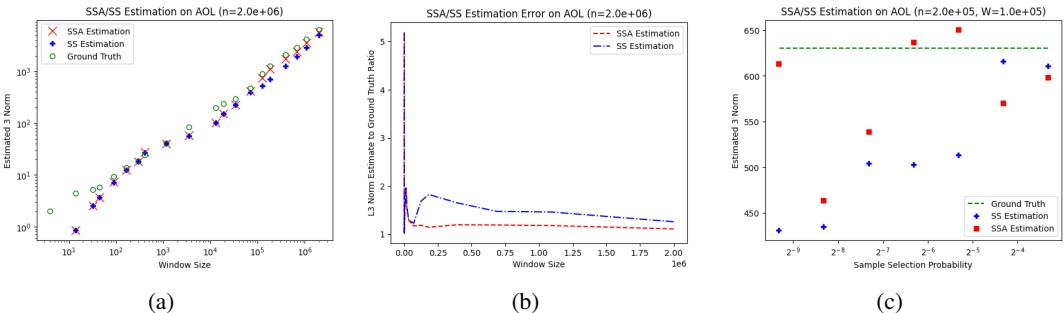

(a)  (b)  (c)

Fig. 3: Experiments for $\ell_3$ estimation on AOL

Figure 3a provides the estimation results from SSA and SS to the actual $\ell_3$ norm over various window sizes for the AOL dataset, a second real-world dataset. The x-axis is converted from timesteps to window sizes and log-scaled for better interpretability; the y-axis is also log scaled. Combined with Figure 3b, we see that SSA is more accurate than SS for $W > 125,000$. However, given its flat error curve and close estimates, SS seems to be a more reliable estimate for the AOL dataset than for the CAIDA dataset. We suspect that SSA is not as advantageous over SS because the AOL dataset is more uniform than the the CAIDA dataset. Nevertheless, SSA remains more accurate compared to SS in this setting. Figure 3c compares the estimation results from SSA and SS to the actual $\ell_3$ norm over various sample selection probabilities. As seen in the figure, SSA provides more accurate estimates of the $\ell_3$ norm than SS for especially small sample selection probabilities. As the probabilities increase, SS benefits from increased sample sizes, ultimately providing better estimates of the $\ell_3$ norm. Cumulatively, SSA provides more accurate estimates over most of the sample selection probabilities, but especially for lower probabilities, indicating that it is more beneficial in low space settings.

## ACKNOWLEDGMENTS

David P. Woodruff is supported in part Office of Naval Research award number N000142112647, and a Simons Investigator Award. Samson Zhou is supported in part by NSF CCF-2335411. Samson Zhou gratefully acknowledges funding provided by the Oak Ridge Associated Universities (ORAU) Ralph E. Powe Junior Faculty Enhancement Award.

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

## A  MISSING DETAILS OF SECTION 3

### A.1  THE SLIDING-WINDOW FRAMEWORK AND THE PROOF OF LEMMA 3.1

Braverman & Ostrovsky (2007) gave a reduction from sliding-window streaming algorithms to the vanilla streaming (approximate) algorithms for $(\alpha, \beta)$-smooth functions. The statement for such reductions is as follows.

**Proposition 1** (Exact algorithms, Theorem 1 of Braverman & Ostrovsky (2007))**.** *Let $f$ be an $(\alpha, \beta)$-smooth function, and let* ALG *be a streaming algorithm that outputs $f(\mathbf{x})$ by the end of the stream, where $\mathbf{x}$ is the frequency vector of the stream. Suppose* ALG *uses $g$ space and performs $h$ operations per streaming update.*

*Then, there exists a sliding-window streaming algorithm* ALG$'$ *that computes a $(1 \pm \alpha)$-approximation of the sliding-windows using $O(\frac{(g + \log n) \cdot \log n}{\beta})$ space and $O(\frac{h \log n}{\beta})$ operations per streaming update.*

Proposition 1 takes *exact and deterministic* streaming algorithms. It turns out that the framework is much more versatile, and we could obtain similar results using *approximate and randomized* streaming algorithms. The new statement is as follows.

**Proposition 2** (Approximate algorithms, Theorem 2 & 3 of Braverman & Ostrovsky (2007))**.** *Let $f$ be an $(\alpha, \beta)$-smooth function, and let* ALG *be a streaming algorithm that outputs $f'(\mathbf{x})$ such that $(1 - \varepsilon) \cdot f(\mathbf{x}) \leqslant f'(\mathbf{x}) \leqslant (1 + \varepsilon) \cdot f(\mathbf{x})$ by the end of the stream with probability at least $1 - \delta$, where $\mathbf{x}$ is the frequency vector of the stream. Suppose* ALG *uses $g(\varepsilon, \delta)$ space and performs $h(\varepsilon, \delta)$ operations per stream update.*

*Then, there exists a sliding-window streaming algorithm* ALG$'$ *that computes a $(1 \pm (\alpha + \varepsilon))$-approximation of the sliding-windows with probability at least $1 - \delta$ using $O\left(\frac{(g(\varepsilon, \delta') + \log n) \cdot \log n}{\beta}\right)$ space and $O\left(\frac{h(\varepsilon, \delta') \log n}{\beta}\right)$ operations per stream update, where $\delta' = \frac{\delta \beta}{\log n}$.*

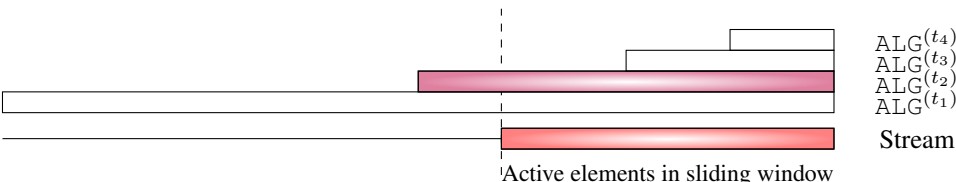

Fig. 4: Example of smooth histogram framework. Here ALG$^{(t_2)}$ and ALG$^{(t_3)}$ sandwich the active elements and are thus good approximations of the sliding window.

At a high level, the algorithm of Braverman & Ostrovsky (2007) uses the idea of *smooth histograms*. The algorithm constructs smooth histograms by running the streaming algorithms with different *starting times* and discarding the redundant copies. An overview of the algorithm is given in Algorithm 1 and an illustration is given in Figure 4.

---

**Algorithm 1.  The algorithm for the framework prescribed in Propositions 1 and 2.**
**Input: a stream of elements with $m$ updates; window size $W$.**
**Input: a streaming algorithm** ALG **with $g(\varepsilon, \delta)$ space and $h(\varepsilon, \delta)$ update time**
**Maintain a set $\mathcal{A}$ of *surviving* copies of** ALG

- For each update $t \in [m]$:
  (1) Initiate a new copy of ALG (call it ALG$^{(t)}$) *starting with the $t$-th update*.

---

> (2) Update all $\texttt{ALG} \in \mathcal{A}$ with $(t, \sigma_t)$.
> (3) **Pruning:**
>   (a) Starting from the algorithm $\texttt{ALG}^{(\ell)} \in \mathcal{A}$ with the smallest index $\ell$.
>   (b) While $\ell < t - 1$:
>     i. Find the largest index $k$ such that $\texttt{ALG}^{(k)} \geqslant (1 - \beta) \cdot \texttt{ALG}^{(\ell)}$.
>     ii. Prune all algorithms in $\mathcal{A}$ with indices $(\ell, k-1]$.
>     iii. Let $\ell \leftarrow k$ and continue the loop.
>     iv. Break the loop if there is no surviving copy between $\ell$ and $t$.
> • Output $\texttt{ALG}^{(t_j)}$, for the largest remaining index $t_j$ with $t_j \leqslant m - W + 1$

**Lemma 3.1.** *Let $f$ be an $(\alpha, \beta)$-smooth function, and let $\texttt{ALG}$ be any learning-augmented streaming algorithm that queries the heavy-hitter oracle $\mathcal{O}$ with the following properties:*

- $\texttt{ALG}$ *outputs $f'(\mathbf{x})$ such that $(1 - \varepsilon) \cdot f(\mathbf{x}) \leqslant f'(\mathbf{x}) \leqslant (1 + \varepsilon) \cdot f(\mathbf{x})$ by the end of the stream with probability at least $1 - \delta$;*

- $\mathcal{O}$ *satisfies the suffix-compatible property as prescribed by Definition 2; and*

- $\texttt{ALG}$ *uses $g(\varepsilon, \delta)$ space and performs $h(\varepsilon, \delta)$ operations per stream update.*

*Then, there exists a sliding-window streaming algorithm $\texttt{ALG}'$ that computes a $(1 \pm (\alpha + \varepsilon))$-approximation of the sliding-windows with probability at least $1 - \delta$ using $O(\frac{(g(\varepsilon, \delta') + \log n) \cdot \log n}{\beta})$ space and $O(\frac{h(\varepsilon, \delta') \log n}{\beta})$ operations per stream update, where $\delta' = \frac{\delta \beta}{\log n}$.*

*Proof.* The correctness of the reductions in Proposition 1 and Proposition 2 relies on the smooth histogram properly approximating the function on the sliding window.

Let $W$ be the window size and $t^* = m - W + 1$ be the starting index of the active window. The goal is to estimate $f$ on the stream suffix starting at $t^*$. The algorithm maintains a set of active instances $\mathcal{A} = \{\texttt{ALG}^{(t_1)}, \texttt{ALG}^{(t_2)}, \ldots, \texttt{ALG}^{(t_k)}\}$ with start times $t_1 < t_2 < \ldots < t_k$.

Because the algorithm only deletes instances that are redundant via pruning, there exist two adjacent instances in $\mathcal{A}$, denoted $\texttt{ALG}^{(t_j)}$ and $\texttt{ALG}^{(t_{j+1})}$, that "sandwich" the true window start time, i.e., $t_j \leqslant t^* < t_{j+1}$.

Let $S^{(t)}$ denote the suffix of the stream starting at time $t$. Observe that $S^{(t_j)} \supseteq S^{(t^*)} \supseteq S^{(t_{j+1})}$. By the hypothesis of suffix-compatibility, the oracle $\mathcal{O}$ provides valid advice to both $\texttt{ALG}^{(t_j)}$ and $\texttt{ALG}^{(t_{j+1})}$. Consequently, these instances correctly output values $v_j \approx f(S^{(t_j)})$ and $v_{j+1} \approx f(S^{(t_{j+1})})$.

The pruning condition in Algorithm 1 ensures that if both instances remain in $\mathcal{A}$, then $v_{j+1} \geqslant (1 - \beta) \cdot v_j$. Since $f$ is $(\alpha, \beta)$-smooth and monotonic, the condition $f(S^{(t_{j+1})}) \geqslant (1 - \beta) f(S^{(t_j)})$ combined with the sandwiching property $S^{(t_j)} \supseteq S^{(t^*)} \supseteq S^{(t_{j+1})}$ implies that the value $v_j$ is a $(1 \pm \alpha)$-approximation of the true window value $f(S^{(t^*)})$.

Therefore, the suffix-compatibility ensures the individual instances are correct, and the smooth histogram ensures the output instance $\texttt{ALG}^{(t_j)}$ is an accurate approximation.

Finally, for the space complexity and the number of operations, we argue that we only keep $O(\frac{\log n}{\beta})$ copies of $\texttt{ALG}$. Note that we delete the copies such that $\texttt{ALG}^{(k)} \geqslant (1 - \beta) \cdot \texttt{ALG}^{(\ell)}$, and the total number of updates can only be $m = \text{poly}(n)$. Therefore, the total number of maintained copies can be at most $O(\log_{\frac{1}{1-\beta}}(m)) = O(\frac{\log n}{\beta})$. We scale $\delta' = \frac{\delta \beta}{\log n}$ to ensure the success probability, and we use $O(\log n)$ space overhead for each copy of the algorithm to maintain the time stamps. Combining the above bounds gives us the desired space and operation complexity.

$\square$

## A.2 MISSING DETAILS OF THE SLIDING-WINDOW $F_p$ ALGORITHM

We provide a brief description of the algorithm as in Algorithm 2, which, in turn, uses the following result as a black-box.

**Proposition 3.** [Alon et al. (1999); Indyk & Woodruff (2005); Andoni et al. (2011)] *There exists a randomized algorithm such that given* $\mathbf{x} \in \mathbb{R}^n$ *as a stream of updates, computes a* $(1 \pm \varepsilon)$-*approximation of* $\|\mathbf{x}\|_p^p$ *with probability at least* $99/100$ *using a space* $O(\frac{n^{1-2/p}}{\varepsilon^{2+4/p}} \cdot \log^2 n)$ *space.*

---

**Algorithm 2. The algorithm for learning-augmented streaming $F_p$ moment.**
**Input: $\mathbf{x}$ given a stream of updates**
**Input: independent copies $\mathtt{ALG}_1$ and $\mathtt{ALG}_2$ for the algorithm in Proposition 3.**

- For each element update on $\mathbf{x}_i$:
  (1) Query whether $\mathbf{x}_i$ is a heavy hitter, i.e., $|\mathbf{x}_i|^p \geqslant \frac{1}{\sqrt{n}} \cdot \|\mathbf{x}\|_p^p$.
  (2) If Yes, use $\mathtt{ALG}_1$ for items with predictions $|\mathbf{x}_i|^p \geqslant \frac{1}{\sqrt{n}} \cdot \|\mathbf{x}\|_p^p$.
  (3) Otherwise, compute the $F_p$ frequency of the non-heavy hitters as follows.
      (a) Sample $\mathbf{x}_i$ with probability $\rho = 1/\sqrt{n}$.
      (b) Let $\tilde{\mathbf{x}}$ be the frequency vector obtained from the sampled non-heavy hitter elements.
      (c) Compute the $F_p$ frequency of $\tilde{\mathbf{x}}$ using $\mathtt{ALG}_2$ and re-weight with $\rho$.
- Summing up the results of $\mathtt{ALG}_1$ and $\mathtt{ALG}_2$ to output.

---

## A.3 RECTANGLE $F_p$ FREQUENCY FOR $p \geqslant 2$

We now move to learning-augmented rectangle $F_p$ frequency algorithms for $p \geqslant 2$. We combine the algorithm statements for deterministic and stochastic oracles as follows.

**Theorem 5.** *There exists a sliding-window streaming algorithm that, given a stream of elements from* $[\Delta]^d$ *in a sliding window, a fixed parameter* $p \geqslant 2$, *and a stochastic suffix-compatible heavy-hitter oracle* $\mathcal{O}$ *with success probability* $1 - \delta$ *(as prescribed by Definition 2), with probability at least* $99/100$ *outputs a* $(1 + \varepsilon)$-*approximation of the rectangle* $F_p$ *frequency in space*

- $O\left(\frac{\Delta^{d(1-1/p)} \cdot p^p \cdot \delta^{1-1/p}}{\varepsilon^{4+p}} \cdot \mathrm{poly}(\frac{p}{\varepsilon}, d, \log \Delta)\right)$ *bits if* $\delta = \Omega(1/\sqrt{n})$.

- $O(\frac{\Delta^{d(1/2-1/p)}}{\varepsilon^{4+p}} \cdot p^p \cdot \mathrm{poly}(\frac{p}{\varepsilon}, d, \log \Delta)$ *bits if* $\delta = O(1/\sqrt{n})$.

*Furthermore, assuming the deterministic oracle, the sliding-window algorithm uses at most* $O(\frac{\Delta^{d(1/2-1/p)}}{\varepsilon^{4+p}} \cdot p^p \cdot \mathrm{poly}(\frac{p}{\varepsilon}, d, \log \Delta))$ *time to process each item.*

*Proof.* The theorem statement before the "furthermore" part follows directly from Theorem 2. In particular, note that the rectangle $F_p$ frequency problem could be framed as $F_p$ frequency with $n \leqslant \Delta^d$, and plugging in the number would immediately lead to the desired space bounds.

For the process time, Jiang et al. (2020) has a $(1 \pm \varepsilon)$-approximate algorithm for the rectangle $F_p$ norm with per-update processing time $O\left(\frac{\Delta^{d(1/2-1/p)}}{\varepsilon^4} \cdot \mathrm{poly}(\frac{p}{\varepsilon}, d, \log \Delta)\right)$ time (without the $p^p/\varepsilon^p$ terms) and success probability $99/100$. We could use the median trick to boost the success probability to $1 - \delta$ with $O(\log(1/\delta))$ space overhead and no time complexity overhead (we could process copies of algorithms in parallel). Therefore, applying Proposition 2 with the same smoothness guarantees as in Lemma 3.3 (rectangle $F_p$ is a sub-family of $\ell_p$ frequencies) leads to the desired $O(\frac{\Delta^{d(1/2-1/p)}}{\varepsilon^{4+p}} \cdot p^p \cdot \mathrm{poly}(\frac{p}{\varepsilon}, d, \log \Delta)$ processing time in the sliding-window model. $\square$

## A.4 $(k, p)$-CASCADED NORMS

We now move the results for the learning-augmented sliding-window $(k, p)$-cascaded norm algorithm. The guarantees of the algorithm are as follows.

**Theorem 6.** *There exists a sliding-window streaming algorithm that, given a $n \times d$ matrix $\mathbf{X}$ represented as a stream of insertions and deletions of the coordinates $\mathbf{X}_{i,j}$, fixed parameters $k \geqslant p \geqslant 2$, and a (deterministic) suffix-compatible heavy-hitter oracle $\mathcal{O}$, with probability at least $99/100$ outputs a $(1 + \varepsilon)$-approximation of the $(k, p)$-cascaded norm in space $O(n^{1 - \frac{1}{k} - \frac{p}{2k}} \cdot d^{\frac{1}{2} - \frac{1}{p}} \cdot \text{poly}(\frac{1}{\varepsilon^k}, \log n))$.*

For any constant choices of $p$, $k$, and $\varepsilon$, our bound asymptotically matches the optimal memory bound for the learning-augmented streaming algorithm. Our algorithm takes advantage of the framework of Jayram & Woodruff (2009) and Jiang et al. (2020) with the smooth histogram framework as in Propositions 1 and 2. The algorithm for streaming learning-augmented cascaded norm is quite involved. As such, we provided a sketch in Algorithm 3, and refer keen readers to Jayram & Woodruff (2009) and Jiang et al. (2020) for more details. In what follows, we use $F_p(\mathbf{X})$ to denote the vector $F_p$ norm of the elements in $\mathbf{X}$.

---

**Algorithm 3. Learning-augmented streaming $(k, p)$-cascaded norm.**
**Input: a $n \times d$ matrix X as the input; parameters $k$ and $p$**
**Input: a heavy hitter oracle predicting whether $|\mathbf{X}_{i,j}|^p \geqslant \|\mathbf{X}\|_p^p / (d^{1/2} \cdot n^{1 - p/2k})$**

- Parameters:
    - $Q = O(n^{1 - 1/k})$ so that $T = (nd \cdot Q)^{1/2} = d^{1/2} \cdot n^{1 - p/2k}$;
    - Levels $\ell \in [O(\log n)]$; layers $t \in [O(\log n / \zeta)]$; $T_\ell = T/2^\ell$.
    - Parameters $\zeta, \eta, \theta, B$ for layering and sampling (as per Jayram & Woodruff (2009)).
- Apply count-sketch type of algorithms (e.g., the algorithm of Proposition 3) during the stream to maintain elements that are sampled by the **level-wise pre-processing** step.
- **Level-wise processing** for level $\ell \in [O(\log n)]$:
    (1) Sample each row with probability $1/2^\ell$; let $\mathbf{X}^{(\ell)}$ be the resulting matrix.
    (2) Divide the entries in $\mathbf{X}^{(\ell)}$ among *layers*: each layer contains the elements with magnitude in $[\zeta\eta^{t-1}, \zeta\eta^t]$.
    (3) A layer $t$ is *contributing* if $|S_t(\mathbf{X}^{(\ell)})|(\zeta\eta^t)^p \geqslant F_p(\mathbf{X}^{(\ell)})/(B\theta)$, where $S_t(\mathbf{X}^{(\ell)})$ are entries in layer $t$.
    (4) Further divide the elements in contributing layers into heavy hitters and non-heavy hitters. This results in contributing layers with entirely heavy hitters vs. non-heavy hitters.
    (5) For each contributing layer $t$:
        (a) If it is formed with non-heavy hitters (light elements) and entries $N_t$ is more than $\beta_t = \theta Q |S_t(\mathbf{X}^{(\ell)})|(\zeta\eta^t)^p/F_p(\mathbf{X}^{(\ell)})$, down sample with rate $\theta Q/T_\ell$.
        (b) Let $j$ be the parameter such that $|S_t(\mathbf{X}^{(\ell)})|/2^j \leqslant \beta_t < |S_t(\mathbf{X}^{(\ell)})|/2^{j-1}$.
        (c) Sample each entry of the layer with rate $1/2^j$ to obtain $\mathbf{Y}_t$ as the resulting matrix.
    (6) Aggregate all $\mathbf{Y}_t$ elements (using the count-sketch algorithm) as the pre-processed vector $\mathbf{Y}^{(\ell)}$.
- Adding up all $\mathbf{Y}^{(\ell)}$ to get $\mathbf{Y}$ and perform "$\ell_p$-sampling" process on $\mathbf{Y}$ in the same manner of Jayram & Woodruff (2009) to obtain $\tilde{\mathbf{Y}}$.
- Compute $F_k(F_p(\tilde{\mathbf{Y}}))$ as the estimation.

---

Jiang et al. (2020) provided the guarantees for Algorithm 3 with the heavy-hitter oracle for vanilla streaming algorithms.

**Lemma A.1** (Jiang et al. (2020)). *Let $\varepsilon > 0$ and $k \geqslant p \geqslant 2$ be given parameters. Furthermore, let $\mathbf{X}$ be an $n \times d$ matrix given as a stream of insertions and deletions of the coordinates $\mathbf{X}_{i,j}$. Then, Algorithm 3 outputs a $(1 \pm \varepsilon)$-approximation of the $(k, p)$-cascaded norm with probability at least $99/100$ using $O(n^{1 - \frac{1}{k} - \frac{p}{2k}} \cdot d^{\frac{1}{2} - \frac{1}{p}} \cdot \text{poly}(\frac{1}{\varepsilon}, \log n))$ space.*

Next, we need to bound the smoothness of the $(k, p)$-cascaded norm. In what follows, we write the $(k, p)$-cascaded norm as a function for "norm of norms", i.e., in the form of $F_k(F_p(\mathbf{X})) :=$

$\left(\sum_{i=1}^{n}\left((\sum_{j=1}^{d}|\mathbf{X}_{i,j}|^{p})^{1/p}\right)^{k}\right)^{1/k}$ for $k \geqslant p \geqslant 2$. Our technical lemma for the smoothness of such functions is as follows.

**Lemma A.2.** *For any constants $k \geqslant p \geqslant 2$, the $(k,p)$-cascaded norm $F_k(F_p(\mathbf{X}))$ is $(\varepsilon, \varepsilon^k/k)$-smooth.*

*Proof.* Let $\mathbf{X}^A$ be a matrix obtained by a suffix of updates of $\mathbf{X}^B$. Furthermore, let $\mathbf{X}^C$ be a common suffix of $\mathbf{X}^A$ and $\mathbf{X}^B$. We also partition the updates in $B = A \cup (B \setminus A)$, and $B \setminus A$ could be empty sets. We further let $\mathbf{X}_{i,:}$ (resp. $\mathbf{X}_{i,:}^{A}$, $\mathbf{X}_{i,:}^{B}$, and $\mathbf{X}_{i,:}^{B \setminus A}$) be the updates of the vector in the $i$-th *row* of $\mathbf{X}$. We lower bound the value of $\left(F_k(F_p(\mathbf{X}^B))\right)^k$ as follows.

$$
\left(F_k(F_p(\mathbf{X}^B))\right)^k = \left|\sum_{i=1}^{n}\left(F_p(\mathbf{X}_{i,:}^{B})\right)^{k}\right|
$$

$$
= \left|\sum_{i=1}^{n}\left(\left(\sum_{j=1}^{d}\left|\mathbf{X}_{i,j}^{A} + \mathbf{X}_{i,j}^{B \setminus A}\right|^{p}\right)^{1/p}\right)^{k}\right|
$$

$$
\geqslant \left|\sum_{i=1}^{n}\left(\left(\sum_{j=1}^{d}|\mathbf{X}_{i,j}^{A}|^{p} + \sum_{j=1}^{d}\left|\mathbf{X}_{i,j}^{B \setminus A}\right|^{p}\right)^{1/p}\right)^{k}\right| \quad \text{(by superadditivity for } p \geqslant 1)
$$

$$
\geqslant \left|\sum_{i=1}^{n}\left(\left(\sum_{j=1}^{d}|\mathbf{X}_{i,j}^{A}|^{p}\right)^{k/p} + \left(\sum_{j=1}^{d}\left|\mathbf{X}_{i,j}^{B \setminus A}\right|^{p}\right)^{k/p}\right)\right|
$$
$$
\text{(by superadditivity for } k \geqslant p)
$$

$$
= \left(F_k(F_p(\mathbf{X}^A))\right)^k + \left(F_k(F_p(\mathbf{X}^{B \setminus A}))\right)^k.
$$

Therefore, for *any* (possibly empty) $\mathbf{X}^{B \setminus A}$, if $F_k(F_p(\mathbf{X}^A))$ and $F_k(F_p(\mathbf{X}^B))$ are $\beta$-close, meaning $F_k(F_p(\mathbf{X}^A)) \geqslant (1-\beta) \cdot F_k(F_p(\mathbf{X}^B))$, we have that

$$
\left(F_k(F_p(\mathbf{X}^{B \setminus A}))\right)^k \leqslant \left(F_k(F_p(\mathbf{X}^B))\right)^k - \left(F_k(F_p(\mathbf{X}^A))\right)^k
$$

$$
\leqslant \left(1 - (1-\beta)^k\right) \cdot \left(F_k(F_p(\mathbf{X}^B))\right)^k
$$

$$
\leqslant k\beta \cdot \left(F_k(F_p(\mathbf{X}^B))\right)^k. \quad \text{(by Bernoulli's inequality)}
$$

Taking the $1/k$-th exponent, we obtain $F_k(F_p(\mathbf{X}^{B \setminus A})) \leqslant (k\beta)^{1/k} \cdot F_k(F_p(\mathbf{X}^B))$.

We now introduce the common suffix $\mathbf{X}^C$, and lower bound the value of $F_k(F_p(\mathbf{X}^{A \cup C}))$ as follows. By Minkowski's inequality, since $k \geqslant p \geqslant 2$, we have that

$$
F_k(F_p(\mathbf{X}^{B \cup C})) \leqslant F_k(F_p(\mathbf{X}^{A \cup C})) + F_k(F_p(\mathbf{X}^{B \setminus A})).
$$

As such, since $\mathbf{X}^C$ is a non-negative frequency matrix, we have that $F_k(F_p(\mathbf{X}^B)) \leqslant F_k(F_p(\mathbf{X}^{B \cup C}))$. Therefore, we could obtain

$$
F_k(F_p(\mathbf{X}^{A \cup C})) \geqslant F_k(F_p(\mathbf{X}^{B \cup C})) - F_k(F_p(\mathbf{X}^{B \setminus A}))
$$

$$
\geqslant F_k(F_p(\mathbf{X}^{B \cup C})) - (k\beta)^{1/k} \cdot F_k(F_p(\mathbf{X}^B))
$$

$$
\geqslant \left(1 - (k\beta)^{1/k}\right) \cdot F_k(F_p(\mathbf{X}^{B \cup C})).
$$

By setting $\beta = \varepsilon^k/k$, we have $(k\beta)^{1/k} = \varepsilon$, which leads to

$$
F_k(F_p(\mathbf{X}^{A \cup C})) \geqslant (1-\varepsilon) \cdot F_k(F_p(\mathbf{X}^{B \cup C})),
$$

which is the desired property for $(\varepsilon, \varepsilon^k/k)$-smoothness. $\qquad \square$

***Finalizing the proof of Theorem 6.*** We again apply Proposition 2 (with Lemma 3.1) to the algorithm of Lemma A.1. By Lemma A.2, the $(k, p)$-cascaded norm is $\left(\varepsilon, \frac{\varepsilon^k}{k}\right)$-smooth.

With the same median trick as we used in the proof of Theorem 1, we could show that we only need $O(\log(1/\delta))$ multiplicative space overhead on the space to ensure Algorithm 3 succeeds with probability at least $1 - \delta$. Therefore, by setting $\beta = \frac{\varepsilon^k}{k}$, we could obtain

$$g(\varepsilon, \delta') = O\left(n^{1 - \frac{1}{k} - \frac{p}{2k}} \cdot d^{\frac{1}{2} - \frac{1}{p}} \cdot \text{poly}\left(\frac{1}{\varepsilon}, \log n\right) \cdot \log(n/\beta)\right)$$

$$\leqslant O\left(n^{1 - \frac{1}{k} - \frac{p}{2k}} \cdot d^{\frac{1}{2} - \frac{1}{p}} \cdot \text{poly}\left(\frac{1}{\varepsilon}, \log n\right)\right).$$

Therefore, the space we need is

$$O\left(g(\varepsilon, \delta')) \cdot \frac{\log n}{\beta}\right) = O\left(n^{1 - \frac{1}{k} - \frac{p}{2k}} \cdot d^{\frac{1}{2} - \frac{1}{p}} \cdot \text{poly}\left(\frac{1}{\varepsilon^k}, \log n\right)\right).$$

This gives the bound as desired by the theorem statement. $\qquad\square$

## B   MISSING DETAILS IN SECTION 4

We give the missing details of Section 4 in this section, including the proof of Theorem 3 and the results. We start with the re-statement of Theorem 3. The algorithm for the framework is shown as in Algorithm 1.

---

**Algorithm 1** Framework for time decay $G$-moment estimation.

---

**Input:** Sketch matrix $\mathbf{A} \in \mathbb{R}^{k \times n}$ for $G$-moment estimation with post-processing function $f(\cdot)$, accuracy parameter $\varepsilon \in (0, 1)$
1: Let $\nu, \eta, m_\nu$ be defined as in Definition 5
2: Maintain a linear sketch with $\mathbf{A}$ for each block $B_i$ of size 1
3: **for** each time $t \in [m]$ **do**
4:      $\mathbf{u} \leftarrow \mathbf{0}^k$
5:      **for** each block $B_i$ **do**
6:          Let $\mathbf{A}\mathbf{v}_i$ be the linear sketch for block $B_i$
7:          Let $t_i$ be the largest timestep in block $B_i$
8:          **if** $t - t_i + 1 \geqslant m_\nu$ **then**
9:             Delete block $B_i$
10:          **else if** all weights in blocks $B_i$ and $B_j$ are within $\sqrt{1 + \eta}$ **then**
11:             Merge blocks $B_i$ and $B_j$
12:          **else**
13:             $w_i' \leftarrow \frac{1}{\sqrt{1 + \eta}} \cdot w(m - t_i + 1)$
14:             $\mathbf{u} \leftarrow \mathbf{u} + w_i' \cdot \mathbf{A}\mathbf{v}_i$
15:      **return** $f(\mathbf{u})$

---

**Theorem 3.** *Given a streaming algorithm that provides a $(1 + \varepsilon)$-approximation to $G$-moment estimation using a linear sketch with $k$ rows, functions $G$ and $w$ that satisfy the $(\varepsilon, \nu, \eta)$-smoothness condition (Definition 5), there exists an algorithm for general time-decay that provides a $(1 + \varepsilon)$-approximation to $G$-moment estimation that uses at most $O\left(\frac{k}{\eta} \log n \log \frac{1}{\nu}\right)$ bits of space.*

*Furthermore, the statement holds true for learning-augmented algorithms as long as the oracle $\mathcal{O}$ is suffix-compatible.*

*Proof.* Consider a fixed $a \in [n]$ and all times $t_{a_1}, \ldots, t_{a_r} \leqslant t$ with updates to $a$. Then the weight of $a$ at time $t$ is $\sum_{j \in [r]} w(t - t_{a_j} + 1)$. Let $w'$ be the weight assigned to time $t$ by the linear sketch. We claim that

$$\frac{1}{\sqrt{1 + \eta}} \cdot \sum_{j \in [r]} w'(t - t_{a_j} + 1) - \nu \leqslant \sum_{j \in [r]} w'(t - t_{a_j} + 1) \leqslant \sum_{j \in [r]} w(t - t_{a_j} + 1).$$

Consider a fixed block $B_i$. Firstly, note that by definition of $n_\eta$ and by construction, the weights of all indices in each block are within a multiplicative factor of $\sqrt{1+\eta}$. All elements in block $B_i$ are assigned weight $w_i'$ to be $\frac{1}{\sqrt{1+\eta}}$ times the weight of the most recent item in $B_i$. Thus, we have $\sqrt{1+\eta} \cdot w(t - t_{a_j} + 1) \leqslant w_i' \leqslant w(t - t_{a_j} + 1)$ for any update $t_{a_j}$ to $a$ within block $B_i$. Finally, for any update $t_{a_j}$ in a block that does not have a sketch must satisfy $t - t_{a_j} + 1 \geqslant n_\nu$. By definition, the weights of all such updates is at most $\nu$. Hence, we have

$$\frac{1}{\sqrt{1+\eta}} \cdot \sum_{j \in [r]} w'(t - t_{a_j} + 1) - \nu \leqslant \sum_{j \in [r]} w'(t - t_{a_j} + 1) \leqslant \sum_{j \in [r]} w(t - t_{a_j} + 1),$$

as desired.

Let $\widehat{G(x_i)}$ be the weight of coordinate $i \in [n]$ implicitly assigned through this process. By definition of $\eta$ and $\nu$, it then follows that $\left(1 - \frac{\varepsilon}{4}\right) \cdot G(x_i) \leqslant \widehat{G(x_i)} \leqslant G(x_i)$. Summing across all $i \in [n]$, we have

$$\sum_{i \in [n]} \left(1 - \frac{\varepsilon}{4}\right) \cdot \sum_{i \in [n]} G(x_i) \leqslant \widehat{G(x_i)} \leqslant \sum_{i \in [n]} G(x_i).$$

Thus, it suffices to obtain a $\left(1 + \frac{\varepsilon}{4}\right)$-approximation to the $G$-moment of the frequency vector weighted by $w'$. Since $\mathbf{A}$ is a linear sketch and $w_i' \cdot \mathbf{v}_i$ is precisely the frequency vector of block $B_i$ weighted by $w'$, then this is exactly what the post-processing function $f$ achieves. Therefore, correctness of the algorithm holds.

It remains to analyze the space complexity. Each linear sketch $\mathbf{A} \cdot \mathbf{v}_i$ uses $O(k \cdot \log n)$ bits of space assuming all weights and frequencies can be represented using $O(\log n)$ bits of space. This can be optimized for specific functions $w(\cdot)$ and $G(\cdot)$, which we shall do for specific applications. In general, observe that we maintain at most three blocks containing weights within a multiplicative factor of $(1 + \eta)$. The smallest weight of an index in a block is at least $\frac{\nu}{1+\eta} \geqslant \frac{\nu}{2}$, while we have $w(1) = 1$ by assumption. Therefore, the number of blocks is at most $3 \log_{(1+\eta)} \frac{2}{\nu}$ since $w$ is non-increasing. Hence, the algorithm uses at most $O\left(\frac{k}{\eta} \log n \log \frac{1}{\nu}\right)$ bits of space.

Finally, the "furthermore" part of the statement regarding the suffix-compatible oracles follows from the same argument as we made in Lemma 3.1. $\qquad\square$

We now present the results for the time-decay models in order. We first consider the polynomial decay model, where we have $w(t) = \frac{1}{t^s}$ for some fixed parameter $s > 0$. For $F_p$ moment estimation, rectangular moment estimation, and cascaded norms, we have that the $G$-moment is still preserved within a factor of $(1 + \varepsilon)$ even when the coordinates are distorted up to a factor of $(1 + O(\varepsilon))$.

**Lemma B.1.** *For the polynomial decay model $w(t) = \frac{1}{t^s}$, it suffices to set $\eta = O(\varepsilon)$ and $\nu = O\left(\frac{\varepsilon}{m^{p-1}}\right)$.*

*Proof.* We provide the proof for the $G$-moment function for the $F_p$ problem, $G(x) = |x|^p$. The statements for cascaded norm and rectangular moment estimation follow analogously.

First, note that for $\eta = \frac{\varepsilon}{100p^2} = O(\varepsilon)$, we have $(1 + \eta)^p - 1 \leqslant \frac{\varepsilon}{4}$. Thus, it follows that for $G(x) = |x|^p$ and for all $x \geqslant 1$, we have

$$G((1+\eta)x) - G(x) \leqslant \frac{\varepsilon}{4} G(x).$$

for $\eta = \frac{\varepsilon}{100p^2} = O(\varepsilon)$, as desired.

Second, note that $G(1) = 1$ and since $p \geqslant 2$, the expression $(x + \nu)^p - x^p$ is maximized at $x = m$. On the other hand, we have for $\nu = \frac{\varepsilon}{100pm^{p-1}}$, $(m + \nu)^p - m^p \leqslant \frac{\varepsilon}{4}$, and thus

$$G(x + \nu) - G(x) \leqslant \frac{\varepsilon}{4} G(1),$$

for all $x \in [1, m]$, as desired. In this case, we can set $m_\nu = \nu^{-2/s}$, which may or may not be larger than $m$, but the latter case does not matter, since there will be no blocks that have been stored for $m + 1$ steps. $\qquad\square$

Recall that in the standard streaming model, $F_p$ moment estimation can be achieved using the following guarantees:

**Proposition 3.** [Alon et al. (1999); Indyk & Woodruff (2005); Andoni et al. (2011)] *There exists a randomized algorithm such that given* $\mathbf{x} \in \mathbb{R}^n$ *as a stream of updates, computes a* $(1 \pm \varepsilon)$-*approximation of* $\|\mathbf{x}\|_p^p$ *with probability at least* $99/100$ *using a space* $O(\frac{n^{1-2/p}}{\varepsilon^{2+4/p}} \cdot \log^2 n)$ *space.*

By applying Theorem 3 to Proposition 3, we have the following algorithm for $F_p$ moment estimation in the polynomial-decay model.

**Theorem 7.** *Given a constant* $p > 2$ *and an accuracy parameter* $\varepsilon \in (0, 1)$*, there exists a one-pass algorithm that outputs a* $(1 + \varepsilon)$-*approximation to the* $F_p$ *moment in the polynomial-decay model that uses* $\widetilde{O}\left(\frac{n^{1-2/p}}{\varepsilon^{2+4/p}}\right)$ *bits of space.*

By comparison, using the approach of Jiang et al. (2020), we have the following guarantees:

**Theorem 8.** *Given a constant* $p > 2$*, an accuracy parameter* $\varepsilon \in (0, 1)$*, and a heavy-hitter oracle* $\mathcal{O}$ *for the data stream, there exists a one-pass algorithm that outputs a* $(1 + \varepsilon)$-*approximation to the* $F_p$ *moment in the polynomial-decay model that uses* $\widetilde{O}\left(\frac{n^{1/2-1/p}}{\varepsilon^{2+4/p}}\right)$ *bits of space.*

Similarly, we can use the following linear sketch for rectangular $F_p$ moment estimation:

**Proposition 4.** *Tirthapura & Woodruff (2012) Given a constant* $p > 2$ *and an accuracy parameter* $\varepsilon \in (0, 1)$*, there exists a one-pass algorithm that uses a linear sketch and outputs a* $(1 + \varepsilon)$-*approximation to the rectangular* $F_p$ *moment in the streaming model that uses* $\widetilde{O}\left(\frac{\Delta^{d(1-2/p)}}{\varepsilon^{2+4/p}}\right)$ *bits of space.*

By applying Theorem 3 to Proposition 4, our framework achieves the following guarantees for rectangular $F_p$ moment estimation in the polynomial-decay model.

**Theorem 9.** *Given a constant* $p > 2$ *and an accuracy parameter* $\varepsilon \in (0, 1)$*, there exists a one-pass algorithm that outputs a* $(1 + \varepsilon)$-*approximation to the rectangular* $F_p$ *moment in the polynomial-decay model that uses* $\widetilde{O}\left(\frac{\Delta^{d(1-2/p)}}{\varepsilon^{2+4/p}}\right)$ *bits of space.*

By comparison, using the approach of Jiang et al. (2020), we have the following guarantees (restated from Section 4):

**Theorem 4.** *Given a constant* $p > 2$*, an accuracy parameter* $\varepsilon \in (0, 1)$*, and a heavy-hitter oracle* $\mathcal{O}$ *for the data stream, there exists a one-pass algorithm that outputs a* $(1 + \varepsilon)$-*approximation to the rectangular* $F_p$ *moment in the polynomial-decay model that uses* $\widetilde{O}\left(\frac{\Delta^{d(1/2-1/p)}}{\varepsilon^{2+4/p}}\right)$ *bits of space.*

Similarly, consider the exponential decay model, where we have $w(t) = s^t$ for some fixed parameter $s \in (0, 1]$. For $F_p$ moment estimation, rectangular moment estimation, and cascaded norms, we have that the $G$-moment is still preserved within a factor of $(1 + \varepsilon)$ even when the coordinates are distorted up to a factor of $(1 + O(\varepsilon))$.

**Lemma B.2.** *For the exponential decay model* $w(t) = s^t$*, it suffices to set* $\eta = O(\varepsilon)$ *and* $\nu = O\left(\frac{\varepsilon}{m}\right)$*.*

*Proof.* We again consider the $G$-moment function for the $F_p$ problem, $G(x) = |x|^p$ as the the proofs for cascaded norm and rectangular moments are similar. First, for $\eta = \frac{\varepsilon}{100p^2} = O(\varepsilon)$, we have $(1 + \eta)^p - 1 \leqslant \frac{\varepsilon}{4}$. Therefore, for $G(x) = |x|^p$ and all $x \geqslant 1$, $G((1 + \eta)x) - G(x) \leqslant \frac{\varepsilon}{4}G(x)$, as required.

Second, recall that $G(1) = 1$. Since $p \geqslant 2$, the quantity $(x + \nu)^p - x^p$ achieves its maximum over $[1, m]$ at $x = m$. For $\nu = \frac{\varepsilon}{100pm}$, we have $(m + \nu)^p - m^p \leqslant \frac{\varepsilon}{4}$. Thus, for every $x \in [1, m]$, $G(x + \nu) - G(x) \leqslant \frac{\varepsilon}{4}G(1)$. Importantly, this value of $\nu$ means we can set $m_\mu = O(\log n)$. $\square$

Therefore, by applying Theorem 3 to the relevant statements, we obtain the following results for $F_p$ moment estimation in the exponential-decay model.

**Theorem 10.** *Given a constant $p > 2$ and an accuracy parameter $\varepsilon \in (0, 1)$, there exists a one-pass algorithm that outputs a $(1 + \varepsilon)$-approximation to the $F_p$ moment in the exponential-decay model that uses $\widetilde{O}\left(\frac{n^{1-2/p}}{\varepsilon^{2+4/p}}\right)$ bits of space.*

**Theorem 11.** *Given a constant $p > 2$, an accuracy parameter $\varepsilon \in (0, 1)$, and a heavy-hitter oracle $\mathcal{O}$ for the data stream, there exists a one-pass algorithm that outputs a $(1 + \varepsilon)$-approximation to the $F_p$ moment in the exponential-decay model that uses $\widetilde{O}\left(\frac{n^{1/2-1/p}}{\varepsilon^{2+4/p}}\right)$ bits of space.*

Similarly, we obtain the following results for rectangular $F_p$ moment estimation in the exponential-decay model.

**Theorem 12.** *Given a constant $p > 2$ and an accuracy parameter $\varepsilon \in (0, 1)$, there exists a one-pass algorithm that outputs a $(1+\varepsilon)$-approximation to the rectangular $F_p$ moment in the exponential-decay model that uses $\widetilde{O}\left(\frac{\Delta^{d(1-2/p)}}{\varepsilon^{2+4/p}}\right)$ bits of space.*

**Theorem 13.** *Given a constant $p > 2$, an accuracy parameter $\varepsilon \in (0, 1)$, and a heavy-hitter oracle $\mathcal{O}$ for the data stream, there exists a one-pass algorithm that outputs a $(1 + \varepsilon)$-approximation to the rectangular $F_p$ moment in the exponential-decay model that uses $\widetilde{O}\left(\frac{\Delta^{d(1/2-1/p)}}{\varepsilon^{2+4/p}}\right)$ bits of space.*

We remark on the main difference between behaviors of our framework in the polynomial-decay model and in the exponential-decay model. Intuitively, the framework will create a logarithmic number of large blocks in the polynomial-decay model, because as the stream progresses, it takes a significantly larger number of updates for the weight to decrease by a factor of $(1 + \eta)$. In contrast, the framework will create a logarithmic number of small blocks in the exponential-decay model, but then the blocks will be truncated after $O(\log n)$ updates.

## C  ADDITIONAL DETAILS FOR THE EXPERIMENTS

When implementing our experiments, we experimentally chose multiple parameters for our augmented and non-augmented algorithms. This section provides details and justifications for these parameters and presents additional experiments.

### C.1  ORACLES & TRAINING

To demonstrate that a heavy-hitter oracle is feasible, we used several oracles in our experiments. All three oracles were used for experiments on the CAIDA dataset, while only the Count-Sketch oracle was used for the other datasets. Each oracle was trained on a data stream prefix and was asked to identify items that would be heavy hitters in the stream suffix.

**Count-Sketch oracle.** For our first oracle, we implemented the well-known Count-Sketch algorithm Charikar et al. (2004) for finding heavy-hitters on a data stream. The prefix sketching results became our heavy hitters for the suffix. For the synthetic and CAIDA datasets, we used a 100K length prefix, repeated the algorithm 5 times, used 300 hashing buckets, and set $\varepsilon = 0.1$. We changed the prefix to 10K for the AOL dataset but maintained the other parameters.

**LLM oracle.** For our Large Language Model (LLM) oracle, we provided the same 100K CAIDA prefix to ChatGPT and Google Gemini and used the following prompt:

> Given this stream subset, predict 26 ip addresses that will occur most frequently in the future data stream

ChatGPT and Google Gemini predicted identical heavy hitters, so we combined their results into a single LLM oracle. Since Count-Sketch identified 26 heavy hitters, we specifically asked for 26 ip addresses to ensure a reasonable comparison between the oracles. The LLM and Count-Sketch algorithms agreed on the identities of 10/26 heavy-hitters.

**LSTM oracle.** For our LSTM oracle, we trained a heavy-hitter predictor on the same 100K CAIDA prefix. The LSTM consisted of an embedding layer that embedded the universe to 32 dimensions, a single LSTM layer with embedding dimension 32 and hidden dimension 64, and a fully connected

output layer. The predictor was trained for 50 epochs with Binary Cross-Entropy (BCE) Loss using Adam Optimizer with learning rate 0.001. The batch size was set to 64.

## C.2   AMS AND LEARNING-AUGMENTED AMS

We implemented Alon et al. (1999)'s algorithm, which we call AMS, as a baseline for $\ell_2$ norm approximation on the CAIDA dataset. We augmented the baseline algorithm with heavy-hitters from the oracles to compare the algorithms' performance. To convert the streaming algorithms into sliding window ones, we tracked multiple instances of each algorithm. Each instance started at a different timestep to account for a different sliding window of the data stream. Relying on Braverman & Ostrovsky (2007), when two instances' $\ell_2$ norm approximation was within a factor of two, we discarded one instance and used the other to approximate the discarded instance's sliding window. We allowed a maximum of 20 algorithm instances. Each instance of the algorithm contained 11 estimates (obtained with different seeds), so we estimated the $\ell_2$ norm as the median of these estimates. For our hash function, we implemented a seeded hashing mechanism in which the hash output is determined by evaluating a low-degree polynomial over the input domain with coefficients derived from a pseudo-random generator. Specifically, we initialized NumPy's default random number generator with a seed value then sampled four integer coefficients over the range $[0, p)$, where p is a large prime (default $2^{31} - 1$). Note that the seed is set to the repetition number. Our input stream item value was coerced to an integer and substituted into a polynomial of degree three, with each term computed modulo $p$ to avoid overflow and maintain arithmetic within a finite field. The polynomial was evaluated incrementally, applying modular reduction at each step, and the final result was mapped into an output space of size 2 via an additional modulo operation. We mapped the final value to $\{-1, 1\}$ by multiplying this output by two and subtracting one.

## C.3   SS AND LEARNING-AUGMENTED SS

For higher order norm estimation, we implemented Indyk & Woodruff (2005)'s selective subsampling (SS) algorithm. Like before, we created an augmented version of the algorithm and compared its performance to the baseline on the CAIDA, AOL, and synthetic datasets. We used the same histogram mechanism to create a sliding window version of both algorithms. We repeated both algorithms 15 times: each timestep instance of the algorithm held 3 sets of 5 estimates (obtained with different seeds) for the same window. We obtained our $\ell_3$ norm estimate by first taking the mean of each of the 5 estimates, then taking the mean of the remaining 3 values. Again, we allowed a maximum of 20 algorithm instances. For our hash function, we took as input a seed and the stream item value, concatenated them into a canonical string representation, and computed a SHA-256 cryptographic hash over this composite input. Note that the seed was obtained by taking the sum of the stream item value and the repetition number. The resulting 256-bit digest was interpreted as a large integer and used to initialize a local instance of Python's pseudorandom number generator. A single uniform variate in the interval $[0, 1)$ was then produced. If the hash value was below our sample selection probability, which we set to $q_{ssa} = \frac{1}{100}$ and $q_{ss} = \frac{1}{10}$, we sampled the stream item, which effectively defined our bucket count.

## C.4   EXPERIMENTS ON ADDITIONAL DATASETS

### C.4.1   SYNTHETIC DATASET

Figure 5a compares the results from the estimation algorithms, SSA and SS, to the actual $\ell_3$ norm over multiple window sizes for our synthetically generated dataset described in Section 5. Additionally, we include "SSA Scaled" and "SS Scaled", which are obtained by scaling the estimates for $W = m$ (the largest window size) by $\frac{W}{m}$ to estimate smaller window sizes. These methods aim to create natural heuristics to transform vanilla streaming algorithms into sliding-window ones. Intuitively, simply rescaling to estimate a smaller window should work well if the distribution remains unchanged over the stream. However, our synthetic data deliberately includes a distribution shift to analyze if our augmented algorithm, SSA, provides benefits when distribution changes occurs. As seen in Figure 5a, the non-augmented algorithms, SS and SS-Scaled, are significantly further from the ground truth than the augmented-algorithms, SSA and SSA-Scaled. This is supported by the error curves in Figure 5b, which show that the gap between the augmented and non-augmented algorithms increases as the window size shrinks, highlighting that an adversarial distribution shift causes the algorithms to lose

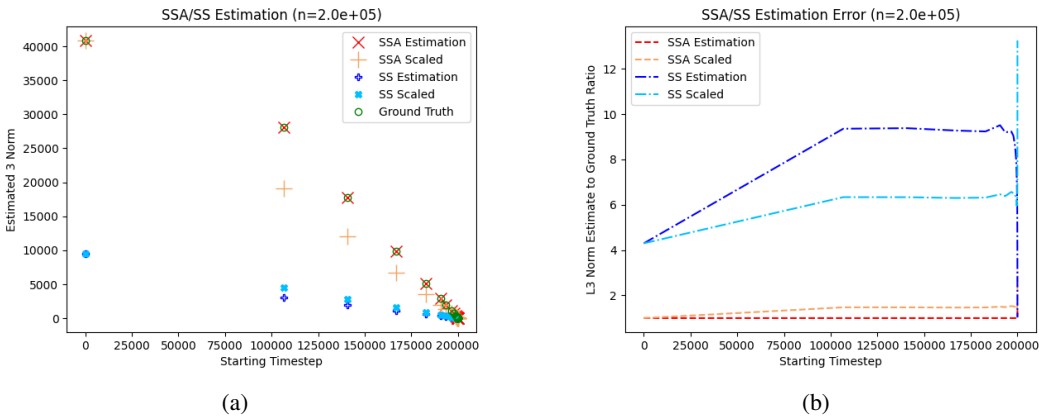

Fig. 5: Experiments for $\ell_3$ estimation on synthetic data

accuracy. Between SSA and SSA-scaled, SSA provides an estimate much closer to the ground truth across window-sizes.

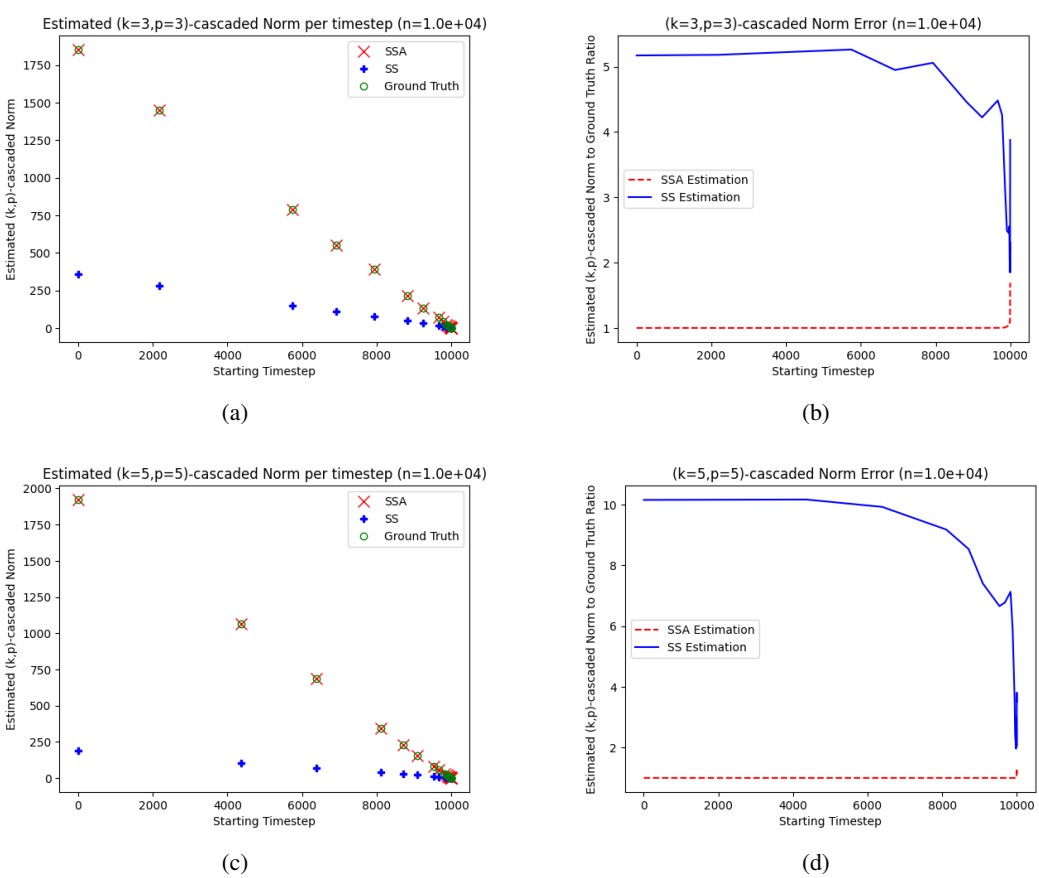

Fig. 6: Experiments for $(k, p)$-cascaded norm estimation on synthetic data

Figure 6 compares the results from the estimation algorithms, SSA and SS, to the actual $(k, p)$-cascaded norm over multiple window sizes for our synthetically generated dataset. Across all window sizes shown in Figure 6a and Figure 6c, SSA, the augmented algorithm, provides a much higher quality estimate than SS. As shown in Figure 6b and Figure 6d, the ratio between the SSA estimate and ground truth value remains nearly constant across all window sizes. Conversely, the SS estimate

seems to degrade exponentially with increased window size. Moreover, compared to its estimate for $(k = 3, p = 3)$, SS provides an estimate that is twice as bad for $(k = 5, p = 5)$-cascaded norms, while SSA remains about equal. This highlights that SSA is relatively stable for higher order norms, while SS degrades more noticeably. Together, the plots suggest that augmenting the baseline algorithm with heavy hitters provides useful information for obtaining higher quality estimates of the $(k, p)$-cascaded norm over various window sizes. In addition to estimation quality, we also monitored the memory usage and running time of the two algorithms. For $(k = 5, p = 5)$-cascaded norm estimation, SSA consumed 68.86 MB of RAM while SS consumed 74.63 MB of RAM, which aligns well with our expectation that the augmented algorithm will consume less memory. The trend is similar for $(k = 3, p = 3)$-cascaded norm estimates as SSA consumed 112.32 MB of RAM, while SS consumed 117.27 MB of RAM. Additionally, for $(k = 5, p = 5)$ SSA ran for 40.3s while SS ran for 63.5s, and for $(k = 3, p = 3)$ SSA ran for 40.1s while SS ran for 61.8s. For both settings, the CountSketch oracle itself used 65.96 MB of RAM. Put together, these results show that SSA can provide higher quality estimates of the $(k, p)$-cascaded norm using less memory than SS while running slightly faster than SS.

## D   THE HEAVY-HITTER ORACLE AND LEARNING THEORY

In this section, we discuss the theoretical aspect of the implementation of the heavy-hitter oracle using the Probably Approximately Correct (PAC) learning framework. The framework helps to demonstrate that a predictor of high quality can be learned efficiently, given that the input instances are from a fixed probability distribution. The discussion of implementing oracles for learning-augmented algorithms enjoys a long history, see, e.g., Izzo et al. (2021); Ergun et al. (2022); Grigorescu et al. (2022); Braverman et al. (2025), and we adapt this framework for the purpose of our heavy-hitter oracles.

Initially, we assume an underlying distribution, denoted as $\mathcal{D}$, from which the input data (frequency vectors of $\mathbf{x}$) is sampled. This setup is standard for solving the frequency estimation problem with or without the learning-augmented oracles. The machine learning model for the oracle would perform well as long as no generalization failure or distribution shift occurs.

Our objective is then to efficiently derive a predictor function $f$ from a given family of possible functions $\mathcal{F}$. The input for any predictor f consists of a frequency vector of $\mathbf{x}$, and the output of the predictor is a vector $\{0, 1\}^n$ indicating whether each $\mathbf{x}_i$ is a heavy hitter. We then introduce a loss function $L : f \times G \to R$, which quantifies the accuracy of a predictor $f$ when applied to a specific input instance $\mathbf{x}$. One could think of $L$ as the function that accounts for the incorrect predictions when compared to the actual heavy-hitter information.

Our goal is to learn the function $f \in \mathcal{F}$ that minimizes the following objective expression:

$$\min_{f \in \mathcal{F}} \mathbb{E}_{\mathbf{x} \sim \mathcal{D}} \left[ L(f(\mathbf{x})) \right]. \tag{1}$$

Let $f^*$ represent an optimal function within the family $\mathcal{F}$, such that $f^* = \operatorname{argmin} \mathbb{E}_{\mathbf{x} \sim \mathcal{D}} \left[ L(f(\mathbf{x})) \right]$ is a function that minimizes the aforementioned objective. Assuming that for every frequency vector $\mathbf{x}$ and every function $f \in \mathcal{F}$, we can compute both $f(\mathbf{x})$ and $L(f(\mathbf{x}))$ in time $T(n)$, we can state the following results using the standard empirical risk minimization method.

**Theorem 14.** *An algorithm exists that utilizes* $\operatorname{poly}\left(T(n), \frac{1}{\varepsilon}\right)$ *samples and outputs a function* $\widehat{f}$ *such that with probability at least* $99/100$*, we have*

$$\mathbb{E}_{\mathbf{x} \sim \mathcal{D}} \left[ L(\hat{f}(\mathbf{x})) \right] \leqslant \min_{f} \mathbb{E}_{\mathbf{x} \sim \mathcal{D}} \left[ L(f(\mathbf{x})) \right] + \varepsilon.$$

In essence, Theorem 14 is a PAC-style result that provides a bound on the number of samples required to achieve a high probability of learning an approximately optimal function.

In what follows, we discuss the proof of Theorem 14 in more detail. We first define the pseudo-dimension for a class of functions, which extends the concept of VC dimension to functions with real-valued outputs.

**Definition 6** (Pseudo-dimension, e.g., Definition 9 in Lucic et al. (2018))**.** Let $\mathcal{X}$ be a ground set, and let $\mathcal{F}$ be a set of functions that map elements from $\mathcal{X}$ to the interval $[0, 1]$. Consider a fixed set

$S = \{x_1, \ldots, x_n\} \subset \mathcal{X}$, a set of real numbers $R = \{r_1, r_2, \cdots, r_n\}$, where each $r_i \in [0, 1]$. Fix any function $f \in \mathcal{F}$, the subset $S_f = \{x_i \in S \mid f(x_i) \geqslant r_i\}$ is known as the induced subset of $S$ (determined by the function $f$ and the real values $R$). We say that the set S along with its associated values R is shattered by $\mathcal{F}$ if the count of distinct induced subsets is $|\{S_f \mid f \in \mathcal{F}\}| = 2^n$. Then, the *pseudo-dimension* of $\mathcal{F}$ is defined as the cardinality of the largest subset of $\mathcal{X}$ that can be shattered by $\mathcal{F}$ (or it is infinite if such a maximum does not exist).

By employing the concept of pseudo-dimension, we can now establish a trade-off between accuracy and sample complexity for empirical risk minimization. Let $\mathcal{H}$ be the class of functions formed by composing functions in $\mathcal{F}$ with $L$; that is, $\mathcal{H} := \{L \circ f : f \in \mathcal{F}\}$. Furthermore, through normalization, we can assume that the range of $L$ is in the range of $[0, 1]$. A well-known generalization bound is given as follows.

**Proposition 5** (Anthony & Bartlett (2002)). *Let $\mathcal{D}$ be a distribution over problem instances in $\mathcal{X}$, and let $\mathcal{H}$ be a class of functions $h : \mathcal{X} \to [0, 1]$ with a pseudo-dimension $d_{\mathcal{H}}$. Consider $t$ independent and identically distributed (i.i.d.) samples $\mathbf{x}_1, \cdots, \mathbf{x}_t$ drawn from $\mathcal{D}$. Then, there exists a universal constant $c_0$ such that for any $\varepsilon > 0$, if $t \geqslant c_0 \cdot \frac{d_{\mathcal{H}}}{\varepsilon^2}$, then for all $h \in \mathcal{H}$, we have that with probability at least $99/100$:*

$$\left| \frac{1}{t} \cdot \sum_{i=1}^{t} h(\mathbf{x}_i) - \mathbb{E}_{\mathbf{x} \sim \mathcal{D}} [h(\mathbf{x})] \right| \leqslant \varepsilon.$$

The following corollary is an immediate consequence derived by applying the triangle inequality on Proposition 5.

**Corollary 15.** *Let $\mathbf{x}_1, \cdots, \mathbf{x}_t$ be a set of independent samples (frequency vectors) drawn from $\mathcal{D}$, and let $\hat{h} \in \mathcal{H}$ be a function that minimizes $\frac{1}{t} \cdot \sum_{i=1}^{t} h(\mathbf{x}_i)$. If the number of samples $t$ is selected as specified in Proposition 5, i.e., $t \geqslant c_0 \cdot \frac{d_{\mathcal{H}}}{\varepsilon^2}$, then with a probability of at least $99/100$, we have*

$$\mathbb{E}_{\mathbf{x} \sim \mathcal{D}} \left[ \hat{h}(\mathbf{x}) \right] \leqslant \mathbb{E}_{\mathbf{x} \sim \mathcal{D}} [h^*(\mathbf{x})] + 2\varepsilon.$$

Finally, we could relate the pseudo-dimension with VC dimension using standard results.

**Lemma D.1** (Pseudo-dimension and VC dimension, Lemma 10 in Lucic et al. (2018)). *For any $h \in \mathcal{H}$, let $B_h$ be the indicator function of the threshold function, i.e., $B_h(x, y) = sgn(h(x) - y)$. Then the pseudo-dimension of $\mathcal{H}$ equals the VC-dimension of the sub-class $B_{\mathcal{H}} = \{B_h \mid h \in \mathcal{H}\}$.*

**Lemma D.2** (Theorem 8.14 in Anthony & Bartlett (2002)). *Let $\tau : \mathbb{R}^a \times \mathbb{R}^b \to \{0, 1\}$, defining the class*

$$\mathcal{T} = \{x \mapsto \tau(\theta, x) : \theta \in \mathbb{R}^a\}.$$

*Assume that any function $\tau$ can be computed by an algorithm that takes as input the pair $(\theta, x) \in \mathbb{R}^a \times \mathbb{R}^b$ and produces the value $\tau(\theta, x)$ after performing no more than $t$ of the following operations:*

- *arithmetic operations $+, -, \times, /$ on real numbers,*

- *comparisons involving $>, \geqslant, <, \leqslant, =$, and outputting the result of such comparisons,*

- *outputting $0$ or $1$.*

*Then, the VC dimension of $\mathcal{T}$ is bounded by $O(a^2 t^2 + t^2 a \log a)$.*

By Lemma D.1 and Lemma D.2, we could straightforwardly bound the VC dimension of the concept class $\mathcal{F}$, which, in turn, bounds the pseudo-dimension of the concept class $\mathcal{F}$. This completes the last peice we need to prove Theorem 14.

*Proof of Theorem 14.* From Lemma D.1 and Lemma D.2, we obtain that the pseudo-dimension of $\mathcal{F}$ is bounded by $O(n^2 \cdot T^2(n))$ by using $a = n$ and $t = T(n)$. This bound could in turn be bounded as $\text{poly}(T(n))$. As such, by Corollary 15, we only need $\text{poly}(T(n))/\varepsilon^2$ samples. We assumed $f(\mathbf{x})$ and $L(f(\mathbf{x}))$ can be computed in time $T(n)$, and applying any poly-time ERM algorithm gives us the total running time of $\text{poly}(T(n), 1/\varepsilon)$, as desired. $\square$

It is important to note that Theorem 14 is a generic framework for learning-augmented oracles. If every function within the family of oracles under consideration can be computed efficiently, then Theorem 14 ensures that a polynomial number of samples will be adequate to learn an oracle that is nearly optimal.

