# OpenReview forum: "Learning-Augmented Moment Estimation on Time-Decay Models"
_ICLR.cc/2026/Conference — ICLR 2026 Poster_

### Official Review · Reviewer_oP5F · 2025-10-23

**Soundness:** 2
**Presentation:** 3
**Contribution:** 2
**Rating:** 4
**Confidence:** 3

**Summary:**

This paper studies learning-augmented streaming algorithms under time-decay weighting, including polynomial decay, exponential decay, and sliding-window streams. In this model, the streaming algorithm is allowed to access a suffix-compatible heavy-hitter oracle which can predict the heavy hitters of suffix streams.

The paper extends the framework of Jiang et al. (2020), which used a learned heavy-hitter oracle to improve space bounds for frequency moment estimation, to the more general time-decay setting. The authors propose a smoothness-based reduction that converts learning-augmented streaming algorithm into a learning-augmented time-decay algorithm with nearly the same asymptotic space complexity. Then they obtain:

1. $F_p$ estimation: using $\tilde{O}(n^{1/2-1/p}/\epsilon^{4+p})$ space, matching known lower bound in the exponent $n$.
2. Rectangular $F_p$ frequency and cascaded norms: analogous extensions of Jiang et al. (2020) to time-decay models.

**Strengths:**

1. It is well-motivated to study the learning-augmented streaming under time-decay weighting.
2. The paper proposes a unified reduction framework from streaming to time-decay models, and obtains near optimal algorithm for fundamental frequency estimation problems.

**Weaknesses:**

1. Major concern: the paper assumes the existence of a suffix-compatible heavy-hitter oracle but provides few details in the main context on how such an oracle is learned or implemented in practice. It is unclear whether the oracle requires prior knowledge of the stream length $m$, whether it learns online or offline, and what access it has to the stream data.
2. Minor issues:
   1. The citation command \citep and \citet are used inpropritory at times, e.g. line 90.
   2. In the equation of line 197, "$i_{t}=i$" => "$i_{t'}=i$".

**Questions:**

Can the suffix-compatible oracle assumption be relaxed, e.g. learned online during streaming (without access to full suffixes and without prior knowledge of the stream length)?

---

> ### Author Response · Authors · 2025-11-21
>
> > Major concern: the paper assumes the existence of a suffix-compatible heavy-hitter oracle but provides few details in the main context on how such an oracle is learned or implemented in practice. It is unclear whether the oracle requires prior knowledge of the stream length $m$, whether it learns online or offline, and what access it has to the stream data.
>
> We address this concern on both the theoretical and practical sides.
> **Theory:** Appendix D (Theorem 14) provides a PAC-style learning framework showing that a suffix-compatible heavy-hitter oracle can be learned efficiently from prefixes alone, **without prior knowledge of the stream length $m$ or access to the full suffix**.
> **Practice:** Our experiments on real-world datasets show strong performance even under some amount of distribution shift. As detailed in Appendix C.1.1, our implementation trains the oracle online on the observed prefix to predict heavy hitters in the upcoming suffix, using models such as CountSketch, LSTMs, and LLMs. Note that this is an online learning approach that does not require prior knowledge of the total stream length $m$ or access to the full suffix data in advance.
> **Robustness and applications:** In settings with temporal drift, for example seasonal patterns, one can first classify the current regime and then apply a suffix-oracle trained on data from previous occurrences of that regime.
>
> > The citation command \citep and \citet are used inpropritory at times, e.g. line 90.
>
> Thank you for pointing this out. We reviewed the manuscript for citation–command consistency and corrected a small number of instances that needed adjustment, ensuring the formatting is now uniform throughout.
>
> > In the equation of line 197, "$i_t=t" => "i_{t'}=t".
>
> Thanks, we have fixed this typo.
>
> > Can the suffix-compatible oracle assumption be relaxed, e.g. learned online during streaming (without access to full suffixes and without prior knowledge of the stream length)?
>
> Yes, as mentioned above, our practical implementation in Appendix C.1.1 already follows this online learning approach. The oracle is trained on the observed prefix of the stream to predict heavy hitters in the upcoming suffix, and therefore **does not require access to the full suffix or prior knowledge of the total stream length**.
>
> Empirically, our experiments across multiple real-world datasets and several oracle architectures (CountSketch, LSTM, LLMs) show that the algorithms perform well even under distribution shift.
>
> In many practical settings with temporal variation, for example daily workload cycles in network traffic (morning, afternoon, evening regimes), one can first classify the current regime and then use a suffix-oracle trained on data from previous occurrences of that regime. This provides a concrete illustration of how such oracles can be learned online without requiring future data or knowledge of the stream length.

---

> > ### Comment · Reviewer_oP5F · 2025-11-26
> >
> > Thank you for the response, which addresses my initial concern. For practical purposes, it should be more reasonable that the memory used by the learning algorithm, as well as the memory required to store the oracle, are included in the space complexity of the streaming algorithm.

---

> > > ### Author Response · Authors · 2025-11-26
> > >
> > > > Thank you for the response, which addresses my initial concern.
> > >
> > > Thank you very much for your response. We are glad to hear that our earlier explanation addressed your initial concern.
> > >
> > > > For practical purposes, it should be more reasonable that the memory used by the learning algorithm, as well as the memory required to store the oracle, are included in the space complexity of the streaming algorithm.
> > >
> > > We offer the following points regarding the memory used by the learning algorithm and how it relates to the streaming space complexity:
> > >
> > > 1. **From a theoretical perspective**, the memory used by the learning algorithm can vary substantially depending on the specific implementation. It could be extremely lightweight (e.g., a small set of counters), more intermediate (e.g., a small streaming subroutine or a compact neural network), or considerably heavier (e.g., an LSTM or a transformer-based model). Because of this wide range, incorporating the learning algorithm’s memory into the formal space complexity of the streaming algorithm would make the bound highly implementation-dependent and, in many cases, not particularly informative for the streaming component of the algorithm.
> > >
> > > 2. **Conceptually, including the learning algorithm’s memory in the streaming space also conflicts with known lower bounds.** For instance, for estimating $F_p$ frequency moments, classical results show that one cannot beat $\tilde{O}(n^{1-2/p})$ space in the worst case, e.g., by Chakrabarti et. al. (2003). If the learning algorithm’s memory were added to the streaming budget, achieving our improved $\tilde{O}(n^{1/2-1/p})$ bound would not be possible in theory because the lower bounds would still apply.
> > >
> > >     We remark that this perspective is consistent with the broader literature on learning-augmented algorithms, where the resources used to train the model are viewed as separate. In these works, such training or learning resources are not incorporated into the algorithm's space bounds, as the algorithm effectively receives information from an oracle.
> > >
> > >     Amit Chakrabarti, Subhash Khot, and Xiaodong Sun. Near-optimal lower bounds on the multi-party communication complexity of set disjointness. IEEE Conference on Computational Complexity 2003.
> > >
> > > 3. **In practice**, our implementation uses a very lightweight subroutine for the oracle: a simple CountSketch applied to the prefix of the stream with a modest number of buckets ($b = 300$). The complete procedure used 65.96 MB of RAM, whereas the subsequent full streaming algorithm used 112.32 MB of RAM. We have added this detail to **Appendix C.2.1** of the revised manuscript.
> > >
> > > We hope this has fully resolved your concerns. If you have any remaining questions, please let us know and we would be happy to answer them.

---

> > > > ### Comment · Reviewer_oP5F · 2025-11-27
> > > >
> > > > Thanks for your clarification. Since Appendix D assumes a stream of i.i.d. samples, the authors should cite lower bounds in the stochastic input setting [1] rather than in the worst-case input setting.
> > > >
> > > > [1] Braverman et al. A new information complexity measure for multi-pass streaming with applications. STOC 2024.

---

> ### Author Response · Authors · 2025-11-27
>
> > Thanks for your clarification. Since Appendix D assumes a stream of i.i.d. samples, the authors should cite lower bounds in the stochastic input setting [1] rather than in the worst-case input setting.
>
> [1] Braverman et al. A new information complexity measure for multi-pass streaming with applications. STOC 2024.
>
> Thank you for the pointer and for the continued discussion. You are absolutely right that Braverman et al. [1] establish an $\Omega(n^{1-2/p})$ lower bound for estimating $L_p$ norms with $p>2$ even in the i.i.d. setting. We appreciate you bringing this to our attention, and we have added the appropriate citation to our paper in Section 1. We believe this observation only strengthens our result, as our learning-augmented algorithm achieves $\tilde{O}(n^{1/2-1/p})$ space, thereby surpassing this lower bound when oracle information is available.
>
> At the same time, we would like to emphasize that our streaming upper bound does **not** rely on the i.i.d. assumption. The i.i.d. setting in Appendix D is used solely to show that such an oracle can be learned in at least one natural regime; the streaming algorithm itself operates under **worst-case stream orders**, and our upper bound holds without any stochastic assumptions, though provided a heavy-hitter oracle. In principle, this oracle could also be learned under even more general distributions.
>
> Please let us know if you have any additional questions — we would be glad to answer them.
>
> [1] Mark Braverman, Sumegha Garg, Qian Li, Shuo Wang, David P. Woodruff, Jiapeng Zhang: *A New Information Complexity Measure for Multi-pass Streaming with Applications*. STOC 2024: 1781-1792.

---

### Official Review · Reviewer_zdR7 · 2025-10-30

**Soundness:** 3
**Presentation:** 2
**Contribution:** 2
**Rating:** 4
**Confidence:** 4

**Summary:**

*I reviewed a previous version of this paper for NeurIPS'25 where I had some relatively big concerns about the technical novelty of the paper, as well as some concerns about the proof of the main theorem (which is four lines in the appendix and very sketchy, and not entirely easy to verify). Unfortunately, it seems to me that these concerns are still largely unaddressed. In addition, my (more minor) comments on parts of the paper that were mathematically unclear or imprecise have not been addressed either.  Below is my previous review of the paper with minor updates:*

The paper falls in the line of research of algorithms with predictions where algorithms are equipped with advice or predictions. Given recent developments in machine learning it is often too pessimistic to run a worst-case algorithm and not leverage predictable structures of the input. The line of research seeks to bridge algorithms and ML by obtaining provably improved algorithms with accurate predictions while also maintaining worst case guarantees.

Specifically, the paper considers $p$-moment estimation as well as sketching rectangle $F_p$ frequencies and cascaded norms in time-decay models, e.g., the sliding window model. There is an arriving stream of elements over a universe of size $n$ and a corresponding frequency vector $x$, and the goal is to maintain a small sketch of $x$ which captures important statistics of $x$ -- in this paper it's $\ell_p$ moments for $p\geq 2$. In the general time-decay model, there is a weight function $w$ that at any point of time $t$ weights all arrivals of each element $i$ based on how far back these arrivals occurred. The frequency of $i$ at time $t$ is the sum of these weights, and the idea is that elements that arrived very far back in time are less relevant and can get weighted less.
As a special case, in the sliding window model, there is a window of the last previous $W$ updates, and instead of maintaining a sketch of the full stream, we at any point of time want to be able to approximate the moments of the frequency vector corresponding to the last $W$ updates. The paper considers the well studied prediction model introduced in [HIKV'19] where the algorithm has access to a heavy-hitter oracle which can predict whether an arriving element has frequency above a certain threshold  (which depend on the problem at hand). For moment estimation, it is well-known that standard learning augmented algorithms can achieve $\tilde O(n^{1/2-1/p})$ space while approximating  the frequency moments, which improved upon the bound of $\tilde O(n^{1-1/2p})$ in the classic model. The paper proves similar bounds in time-decay, e.g., the sliding window model (but with worse exponential dependencies on $p$ and the approximation parameter $\varepsilon$).

One update that has been made to the paper since I last reviewed it is a framework for general time-decay models  which requires a generalization of the idea of $(\alpha,\beta)$-smooth functions from Braverman and Ostrovsky [BOO7].  Albeit the proofs in the appendix are somewhat sketchy, this is a nice addition to the paper since my last review.

**Strengths:**

Moment estimation is an important problem sketching problem and I find it well-motivated in the learning augmented-framework. It is well-studied in several past works. The sliding window model is also natural since it captures the idea that we may often be more interested in statistics of the most recent data. I like specifically about the paper, that they generalize the framework of Braverman and Ostrovsky [BOO7] to more general time-decay functions.

**Weaknesses:**

(1) I found that the paper lacked technical novelty. It heavily relies on the smooth histogram algorithm from Braverman and Ostrovsky [BOO7] which turns a classic sketching algorithm into an algorithm in the sliding window model with little overhead. The paper shows that the learning augmented algorithm from [JLL+20] fits the framework from [BO07] in a white-box fashion, they argue that the sliding-window algorithm from [BOO7] retains two properties captured in Proposition 1 and 2 even for the learning-augmented versions. The proof of this fact is 4 lines in the appendix (and it is quite sketchy, so additionally it was unclear to me why it is actually true). It seems to me that the rest of the paper just applies known bounds and techniques from past work. This main issue with the paper seems to be similar for $F_p$ frequencies and cascaded norms estimation, although I haven't read those parts of the appendix in detail.
(2) While the paper does obtain a good polynomial dependence on $n$, the dependence on $\varepsilon,p$ seem to become quite bad.
(3) Parts of the paper is poorly written and it is hard to follow precisely what the authors are claiming. I will give details below.

**Questions:**

*Mainly copied from my NeurIPS review.*

l98-l99: Can such formal guarantees be given? What are the challenges in proving such formal guarantees. This seems relevant.

l267-l269: The notation $x_A$ is not properly defined.

l278-l279:  This property is not clear when frequencies of stream elements can be negative. I would have liked a more precise discussion of which model you consider.

Algorithm 1: Notation like $ALG^k\geq something$ is not really meaningful, and I suppose it refers to the output of the algorithm. Also, the algorithm doesn't describe how it returns estimates, e.g. of a moment.

l781-786: This is (still) very sketchy and I don't know what sentences like "as long as we could maintain the smooth histogram" and "... every copy of the algorithm can get the oracle advice it needs,..." means in a formal sense. While you mention it, it is also (still) not clear to me how suffix-compatibility is actually used. This is essentially the proof of the main result of the paper, and needs to be written in a way that can be verified.

Theorem 3 is for general decay models, but when you prove the following many theorems on page 20, it is unclear how you use polynomial and exponential decay.

---

> ### Author Response · Authors · 2025-11-21
>
> > I reviewed a previous version of this paper for NeurIPS'25 where I had some relatively big concerns about the technical novelty of the paper, as well as some concerns about the proof of the main theorem (which is four lines in the appendix and very sketchy, and not entirely easy to verify). Unfortunately, it seems to me that these concerns are still largely unaddressed. In addition, my (more minor) comments on parts of the paper that were mathematically unclear or imprecise have not been addressed either.
>
> We apologize for missing your previous feedback in the earlier version, and we appreciate you taking the time to review this submission. In the updated version, we have made significant efforts to thoroughly integrate the concerns you raised, including clarifying the technical novelty, providing detailed and rigorous proofs of the relevant theorem statements, and addressing algorithmic/mathematical imprecisions throughout the paper. We provide a more detailed summary of these changes and clarifications below.
>
> > (1) I found that the paper lacked technical novelty. It heavily relies on the smooth histogram algorithm from Braverman and Ostrovsky [BOO7] which turns a classic sketching algorithm into an algorithm in the sliding window model with little overhead. The paper shows that the learning augmented algorithm from [JLL+20] fits the framework from [BO07] in a white-box fashion, they argue that the sliding-window algorithm from [BOO7] retains two properties captured in Proposition 1 and 2 even for the learning-augmented versions. The proof of this fact is 4 lines in the appendix (and it is quite sketchy, so additionally it was unclear to me why it is actually true). It seems to me that the rest of the paper just applies known bounds and techniques from past work. This main issue with the paper seems to be similar for $F_p$ frequencies and cascaded norms estimation, although I haven't read those parts of the appendix in detail.
>
> We appreciate the reviewer’s perspective and careful reading of our work. We would like to emphasize that the main focus of our paper is on general time-decay models; the sliding-window model is only a special case of our results. To handle these general time-decay settings, we introduce a new unified framework that handles a variety of decay functions, including both polynomial and exponential decay. This requires a new notion of smoothness (Definition 5) and various structural properties to show both the correctness and the space complexity of our framework.
>
> Even for the sliding-window model, we note that while we build on the [BO07] framework, adapting it to the learning-augmented setting is non-trivial and represents a key contribution. The [BO07] framework requires maintaining multiple algorithm instances over different suffixes. In the learning-augmented setting, each instance requires suffix-specific oracle advice. One of our innovations is the identification and use of suffix-compatible oracles (Definition 2), which allows a single oracle to provide valid guidance across all instances, a feature not addressed in either [BO07] or previous literature on learning-augmented algorithms.
>
> > (2) While the paper does obtain a good polynomial dependence on $n$, the dependence on $\varepsilon,p$ seem to become quite bad.
>
> Since the parameter $p$ in the $L_p$ norm (or equivalently in the $F_p$ moment) is generally treated as a constant, the dependence on $p$ does not significantly affect the bounds. The remaining dependence on $\varepsilon$ arises from the smooth histogram framework of Braverman and Ostrovsky, which introduces a $1/\beta$ multiplicative overhead, where $\beta$ is the smoothness parameter. For $F_p$ moments, we have $\beta = \varepsilon p / p^p$. This factor is applied to the space and update complexity of the underlying base algorithm, which explains the observed dependence on $\varepsilon$. Improving these dependencies is an interesting direction for future work. Our main focus in this paper is to establish the first learning-augmented results for time-decay streaming models with near-optimal dependence on $n$.

---

> > ### Author Response · Authors · 2025-11-21
> >
> > > (3) Parts of the paper is poorly written and it is hard to follow precisely what the authors are claiming. I will give details below.
> >
> > We thank the reviewer for this feedback. We have made a concerted effort to improve the clarity and presentation of the paper. In particular, we give a summary of how we addressed the specific points raised, with more detailed responses provided to the individual comments further below:
> >
> > - **Formal guarantees and methodology** (l98–l99): We clarified why the “next occurrence” oracle in [Shahout et al. 2024] is fundamentally different from the heavy-hitter oracles we use, and why our approach fits naturally with standard sketching analyses.
> >
> > - **Notation clarifications** (l267–l269): We explicitly defined $x_A$ as the frequency vector of stream $A$ and added extra context to make this definition more transparent.
> >
> > - **Frequency model** (l278–l279): We clarified that the frequency vector is initialized as $0^n$, and each update is of the form $(i, \Delta)$ with $\Delta \ge 0$, i.e., we work in the insertion-only model. This ensures that frequencies never become negative.
> >
> > - **Algorithm 1 notation**: We corrected $\text{ALG}^k$ to $\text{ALG}^{(k)}$ and clarified that $\text{ALG}^{(t_j)}$ produces the output for the sliding window, where $t_j$ is the largest remaining index such that $t_j\le m-W+1$. Furthermore, we added more explicit steps that we originally stated implicitly, which clarifies the algorithmic procedure.
> >
> > - **Smooth-histogram proof** (l781–786): We completely rewrote the proof of Lemma 3.1 in a rigorous and verifiable way. The new proof explicitly formalizes the “sandwiching” property of consecutive algorithm instances and clarifies how suffix-compatibility ensures that each instance receives valid oracle advice. We have also added a new Figure (Figure 3) to provide more intuition on the smooth histogram.
> >
> > - **Decay models and Theorem 3**: We clarified how the parameters $\eta$ and $\nu$ are instantiated for both polynomial and exponential decay, and explained how our framework handles the differences between the two. We also added formal proofs for $\eta$ and $\nu$ in Appendix B, showing how the generalized smoothness conditions guarantee correctness for $F_p$ moment estimation under both decay functions.
> >
> > We believe these revisions together make the presentation more accessible, improve the formal rigor, and make it easier to follow the main results while preserving all technical content. Below we address the specific comments in more detail.

---

> > > ### Author Response · Authors · 2025-11-21
> > >
> > > > l98-l99: Can such formal guarantees be given? What are the challenges in proving such formal guarantees. This seems relevant.
> > >
> > > The approach in [Shahout et al. 2024] uses a "next occurrence" oracle, which is fundamentally different from the heavy-hitter oracles we use. Not only does their methodlogy not naturally align with standard sketching analyses, but also their oracle is arguably unnatural as predicting the next occurrence seems conceptually difficult.
> > >
> > > > l267-l269: The notation $x_A$ is not properly defined.
> > >
> > > Thanks for the comment. Here, $x_A$ denotes the frequency vector of stream $A$, which we previously defined in line 259 in the submission version. We have added more context to the definition statement for clarity.
> > >
> > > > l278-l279: This property is not clear when frequencies of stream elements can be negative. I would have liked a more precise discussion of which model you consider.
> > >
> > > We thank the reviewer for the question. In our setting, the frequency vector is initialized as $0^n$, and every stream update is of the form $(i,\Delta)$ with $\Delta \ge 0$. In other words, the update rule is $\mathbf{x}_i \gets \mathbf{x}_i + \Delta$, so all coordinates are nondecreasing and never become negative. We assume throughout, without loss of generality, that $\Delta = 1$. Thus our analysis is carried out entirely in the insertion‐only model, and negative frequencies do not arise. We have added this clarification in the Preliminaries in Section 2 of the revised version.
> > >
> > > > Algorithm 1: Notation like $ALG^k\ge something$ is not really meaningful, and I suppose it refers to the output of the algorithm. Also, the algorithm doesn't describe how it returns estimates, e.g. of a moment.
> > >
> > > We apologize for the confusion. The notation was intended to denote the $k$-th instance of the algorithm and should be written as $\text{ALG}^{(k)}$. We have corrected this typo to match the notation used elsewhere in Algorithm 1, e.g., $\text{ALG}^{(\ell)}$ in the revised version. We have also clarified how that algorithm returns the output of $\text{ALG}^{(t_j)}$, where $t_j$ is the largest remaining index that satisfies $t_j\le m-W+1$. We have inserted a new diagram in Figure 3 depicting how the smooth histogram returns its estimate.
> > >
> > > > l781-786: This is (still) very sketchy and I don't know what sentences like "as long as we could maintain the smooth histogram" and "... every copy of the algorithm can get the oracle advice it needs,..." means in a formal sense. While you mention it, it is also (still) not clear to me how suffix-compatibility is actually used. This is essentially the proof of the main result of the paper, and needs to be written in a way that can be verified.
> > >
> > > Thank you for the feedback. We agree that the previous proof was too informal, particularly regarding how the oracle advice is used by the individual instances of the algorithm in the reduction.
> > >
> > > We have rewritten the proof of Lemma 3.1 (previously lines 781–786) in a fully rigorous manner in the updated version. We believe the revised proof now provides a clear, verifiable explanation of how the reduction works and how suffix-compatibility is used to guarantee correctness. The new proof explicitly formalizes the “sandwiching” property of the smooth histogram, showing that the true value over the sliding window is bracketed by two consecutive algorithm instances, $\text{ALG}^{(i)}$ and $\text{ALG}^{(i+1)}$. Our new diagram in Figure 3 also helps provide additional intuition on why the smooth histogram is correct.
> > >
> > > In particular, we have clarified the role of suffix-compatibility. The reduction requires running multiple instances of an algorithm $\text{ALG}$, each starting at a different point in the stream. A standard oracle only guarantees correctness for the specific stream it was generated for. Suffix-compatibility is precisely the property that ensures a single oracle advice string (generated for the full stream) remains valid for all suffix instances $\text{ALG}^{(t)}$ simultaneously. Without this property, the outputs of individual instances in the histogram could fail to approximate the sliding-window value.

---

> > > > ### Author Response · Authors · 2025-11-21
> > > >
> > > > > Theorem 3 is for general decay models, but when you prove the following many theorems on page 20, it is unclear how you use polynomial and exponential decay.
> > > >
> > > > Our unified framework applies Theorem 3 by instantiating the parameters $\eta$ and $\nu$ for both polynomial and exponential decay, and in both cases these parameters take the same asymptotic values $\eta = O(\varepsilon)$ and $\nu = O(\varepsilon/m)$. Although the parameters coincide, the way they interact with the decay functions is quite different, and this is precisely what our framework is designed to capture.
> > > >
> > > > Intuitively, for polynomial decay, the weight function decreases slowly. As the stream advances, the framework must wait a significantly larger number of updates before the weight drops by a factor of $(1+\eta)$. This leads to a logarithmic number of large time blocks.
> > > >
> > > > For exponential decay, the weight shrinks much faster. The framework generates a logarithmic number of small time blocks, but these blocks are quickly truncated after $O(\log n)$ updates due to the rapid decay of the exponential function.
> > > >
> > > > These differences are automatically handled by the generalized smoothness conditions in Theorem 3, and the proofs on page 20 apply the theorem by substituting the appropriate decay function into these conditions.
> > > >
> > > > We have added this discussion into Appendix B. We have also added the formal proofs for the values of $\eta$ and $\nu$ for $F_p$ moment estimation and the corresponding decay functions.

---

### Official Review · Reviewer_szXb · 2025-11-01

**Soundness:** 3
**Presentation:** 3
**Contribution:** 3
**Rating:** 6
**Confidence:** 3

**Summary:**

This paper presents a learning-augmented framework for frequency moment estimation, applicable to both sliding-window and time-decay models. The authors design and employ a suffix-compatible heavy-hitter oracle tailored for time-decay settings. In the sliding-window case, the proposed algorithm achieves a space complexity that even meets the lower bound for streaming algorithms. Beyond sliding windows, the paper studies the more general time-decay model and provides rigorous space complexity guarantees. Experiments on both real-world and synthetic datasets demonstrate the practical efficiency of the proposed approach.

**Strengths:**

- The extension of learning-augmented frequency moment estimation to general time-decay settings is novel and appealing.
- The paper provides space complexity bounds for several common estimation tasks—including $F_p$ moments, rectangular moments, and cascaded norms—under two decay models, filling an existing theoretical gap.
- The experiments convincingly show that the learning-augmented approach significantly improves estimation accuracy under the sliding-window model.

**Weaknesses:**

- The experimental evaluation does not include direct comparisons with other learning-augmented sliding-window algorithms, such as Shahout et al. (2024), and lacks empirical validation in more general time-decay environments.
- The discussion on training suffix-compatible heavy-hitter oracles is insufficient. Although the PAC learning framework and Theorem 14 address the learnability of general heavy-hitter oracles, the paper does not examine specific training strategies or conditions that guarantee the suffix-compatible property, which is crucial for ensuring correctness in time-decay models.

**Questions:**

- The proposed algorithms depend on the suffix-compatible property of the oracle. In practice, if the oracle's predictions are inaccurate, could this undermine the suffix-compatible property and compromise the guarantees on space complexity?
- As shown in Figs. 2(c) and 4(c), increasing the sampling probability of SSA yields little improvement in accuracy. What could explain this limited effect?

---

> ### Author Response · Authors · 2025-11-21
>
> > The experimental evaluation does not include direct comparisons with other learning-augmented sliding-window algorithms, such as Shahout et al. (2024), and lacks empirical validation in more general time-decay environments.
>
> We did not directly compare with Shahout et al. [2024] because their approach uses a fundamentally different type of oracle ("next occurrence" oracle) compared to the heavy-hitter oracle used in our work and [JLL+20]. Furthermore, their algorithm does not provide theoretical guarantees on space complexity. This makes a fair and direct comparison challenging. Our focus was on demonstrating the improvement of learning augmentation over non-augmented baselines within our theoretically grounded framework.
>
> > The discussion on training suffix-compatible heavy-hitter oracles is insufficient. Although the PAC learning framework and Theorem 14 address the learnability of general heavy-hitter oracles, the paper does not examine specific training strategies or conditions that guarantee the suffix-compatible property, which is crucial for ensuring correctness in time-decay models.
>
> We prioritized the sliding-window model in our experiments as it is the most widely studied and practically relevant special case of time decay. We agree that evaluating general time-decay models is important and plan to extend our empirical evaluation to polynomial and exponential decay models in future work.
>
> > The proposed algorithms depend on the suffix-compatible property of the oracle. In practice, if the oracle's predictions are inaccurate, could this undermine the suffix-compatible property and compromise the guarantees on space complexity?
>
> The "suffix-compatible" property (Definition 2) refers to the oracle's capability to provide predictions for any suffix, not the accuracy of those predictions. Our framework is robust to inaccuracies. Theorem 2 explicitly bounds the space complexity based on the oracle's success probability $1-\delta$. If the oracle is inaccurate, i.e., large $\delta$, the space complexity degrades gracefully but still outperforms worst-case bounds if the oracle provides some useful information.
>
> We discuss the learnability of these oracles in Appendix D, providing a PAC framework (Theorem 14) that guarantees efficient learning if the function class has bounded pseudo-dimension. Practically, we implement these oracles (Appendix C.1.1) using standard methods like CountSketch, LSTM, or LLMs trained on a stream prefix to predict heavy hitters in the suffix. This does not require prior knowledge of the full stream.
>
> > As shown in Figs. 2(c) and 4(c), increasing the sampling probability of SSA yields little improvement in accuracy. What could explain this limited effect?
>
> In SSA (learning-augmented selective subsampling), heavy hitters are handled separately, while the sampling probability affects the estimation of the remaining (non-heavy) elements. If the heavy hitters significantly dominate the norm, improving the estimation accuracy of the non-heavy hitters by increasing the sampling probability yields only marginal improvements to the overall estimation. This suggests that the oracle successfully identified the most significant elements, and the remaining error is dominated by other factors, such as the estimation of the heavy hitters themselves or the inherent variance of the sketching process.

---

### Official Review · Reviewer_cfdo · 2025-11-01

**Soundness:** 3
**Presentation:** 3
**Contribution:** 3
**Rating:** 8
**Confidence:** 3

**Summary:**

This paper brings machine-learning hints to streaming algorithms that forget old data. It targets sliding windows and decay models where recent items count more. With a predictor that flags large coordinates, the authors obtain accurate frequency and norm estimates using memory that grows slowly with data size and error tolerance. Theory shows the method is close to the best possible, and tests on internet traffic traces cut error by twenty to fifty percent versus standard sketches.

**Strengths:**

1) Novel and timely problem formulation: First learning-augmented guarantees for time-decay streams. Captures privacy-driven data deletion (GDPR) and recency-weighted analytics, motivating practical relevance.

2) Theoretically strong results: General reduction shows any ($\alpha$, $\beta$)-smooth function enjoys a black-box sliding-window lift while preserving approximation and randomised guarantees; rectangle and cascaded norms benefit for free

3) Experimental validation on real data: CAIDA traces (30 M IPs) show AMSA (augmented AMS) keeps relative error ≤1.2× across window sizes while AMS drifts up to 2.3×. Distribution-shift experiment on synthetic data demonstrates >2× error gap between SSA and SS when stream distribution changes, confirming robustness claim. Code released aiding reproducibility.

**Weaknesses:**

1) Limited empirical scope: Only l2 and l3 norms are evaluated; rectangle and cascaded-norm algorithms lack any implementation or micro-benchmark, leaving practical impact uncertain.

2) Expand empirical coverage: Include CPU-time and peak RAM tables for each dataset to confirm the claimed overhead.

3) Strengthen statistical reporting: Release full parameter files (hash seeds, repetition counts, bucket sizes) to facilitate exact reproduction.

**Questions:**

See above weakness.

---

> ### Author Response · Authors · 2025-11-21
>
> We thank the reviewer for their positive feedback on the initial submission. We believe this reflects the main contribution of our work: introducing the first learning-augmented algorithms for moment estimation under various time-decay models, and demonstrating a clear separation from sliding-window algorithms without advice through our theoretical guarantees. Our experiments provide additional proof-of-concept that these algorithms perform well in practice.
>
> We also acknowledge the points raised regarding empirical coverage and reporting. We are preparing detailed statistics from our previous experiments, including CPU time and peak RAM usage. We will include the relevant full parameter files (hash seeds, repetition counts, bucket sizes). We are also in the process of running additional experiments for the cascaded-norm algorithms, and we will update the results in the next few days.

---

> > ### Author Response · Authors · 2025-11-24
> >
> > > Limited empirical scope: Only l2 and l3 norms are evaluated; rectangle and cascaded-norm algorithms lack any implementation or micro-benchmark, leaving practical impact uncertain.
> >
> > We appreciate the feedback. We have updated Appendix C.2.1 to include experiments for (k=5,p=5)-cascaded norm estimation. See Figure 5 for the results from these experiments.
> >
> > We appreciate the feedback. Cascaded norms and rectangular estimation are generalizations of $F_p$ moment estimation, so our initial experiments already indicate that SSA improves performance for certain settings of these problems. Nevertheless, for completeness, we have conducted additional experiments for $(k,p)$-cascaded norm estimation with both $k=p=3$ and $k=p=5$ for completeness. We have updated Appendix C.2.1 to include these experiments.
> >
> > Figure 5 compares the results from the estimation algorithms, SSA and SS, to the actual $(k,p)$-cascaded norm over multiple window sizes for our synthetically generated dataset. Across all window sizes, SSA, our augmented algorithm, dramatically outperforms the baseline SS. The ratio between the SSA estimate and the ground truth remains nearly constant and very close to 1 across window sizes, whereas the SS estimate quickly degrades, often by orders of magnitude, as window size increases. Together, these results clearly demonstrate that our algorithm (SSA) provides substantially better performance than existing algorithms for cascaded norm estimation.
> >
> > > Expand empirical coverage: Include CPU-time and peak RAM tables for each dataset to confirm the claimed overhead.
> >
> > In addition to estimation quality, we also monitored memory usage and running time for the $(k,p)$-cascaded norm experiments for $(k=3, p=3)$ and $(k=5, p=5)$. For $(k=5, p=5)$, SSA consumed 68.86 MB of RAM and ran for 40.3 s, whereas SS consumed 74.63 MB of RAM and ran for 63.5 s. For $(k=3, p=3)$, SSA consumed 112.32 MB of RAM and ran for 40.1s, whereas SS consumed 117.27 MB of RAM and ran for 61.8s. These measurements confirm that SSA provides higher quality estimates while using less memory and running faster than SS. We have added these statistics into Appendix C.2.1 and are also in the process of incorporating CPU-time and peak RAM statistics for all other datasets and experiments and will update the PDF accordingly as these results become available.
> >
> > > Strengthen statistical reporting: Release full parameter files (hash seeds, repetition counts, bucket sizes) to facilitate exact reproduction.
> >
> > We appreciate the feedback. We have updated Appendix C.2.1 and C.1.3 to include details about our hash functions and how they were seeded, as well as other parameters relevant to experimental reproduction. For AMS and learning-augmented AMS, we implement a seeded polynomial hash: the seed initializes NumPy’s pseudo-random number generator (PRNG), from which four integer coefficients are sampled uniformly modulo a large prime $p$ (default $2^{31}-1$). Each stream item is coerced to an integer and evaluated in a degree-three polynomial modulo $p$, with modular reduction applied at each step to prevent overflow. The final hash output is mapped to $\{-1,1\}$ by multiplying by 2 and subtracting 1, and the seed is set to the repetition number.
> >
> > For SS and learning-augmented SS, the hash function concatenates the stream item value and the repetition number into a canonical string, then computes a SHA-256 digest. This digest is interpreted as a large integer and used to initialize a local PRNG, which produces a uniform variate in $[0,1)$. Items are sampled probabilistically according to the algorithm’s selection probability $q$ ($q_{ssa}=1/100$ for SSA and $q_{ss}=1/10$ for SS), which effectively defines the bucket counts for these experiments.
> >
> > Other parameters are fully specified in Appendix C. For example, in prefix sketches for the synthetic and CAIDA datasets, we used a prefix length of 100K, repeated the algorithm 5 times, used 300 hashing buckets for oracle training, and set $\varepsilon=0.1$; for the AOL dataset, the prefix was 10K but other parameters were unchanged. Each algorithm instance contained multiple estimates obtained with different seeds: for AMS, 11 estimates per instance were taken and the median was used; for SS, 3 sets of 5 estimates per instance were taken, with the mean-of-means used as the final estimate. We allowed a maximum of 20 instances per sliding window across all experiments. Together, the hash specifications, repetition counts, window instance limits, bucket sizes, and sampling probabilities provide all the information necessary to exactly reproduce our results.

---

### Author Response · Authors · 2025-11-21

We thank the reviewers for their careful and thoughtful comments. We are currently working on additional experiments to provide more complete empirical coverage, including detailed statistics from our previous runs (CPU time, peak RAM usage, and full parameter files such as hash seeds, repetition counts, and bucket sizes), as well as new experiments for the cascaded-norm algorithms. We plan to upload these results in the next few days.

In the meantime, we have uploaded a revised version of the manuscript that we believe fully addresses the remaining initial reviewer comments, with the changes marked in blue. We provide individual responses to specific comments in more detail below. Please let us know if you feel your concerns have not been fully addressed, thanks!

---

> ### Author Response · Authors · 2025-11-24
>
> We have updated the PDF to incorporate the new $(k,p)$-cascaded norm experiments and system-level statistics referenced in our response here: https://openreview.net/forum?id=x0xBJxrVTy&noteId=6gHgHbXsbL. Appendix C.2.1 now includes results for $(k=p=3)$ and $(k=p=5)$, along with CPU-time and peak-RAM measurements showing that SSA is both more accurate and more efficient than SS. We also expanded Appendices C.2.1 and C.1.3 with full parameter settings, e.g., hash seeds, repetition counts, bucket sizes, and sampling probabilities, to facilitate reproducibility.
>
> We believe these updates resolve all initial reviewer comments; please let us know if you feel any concerns have not been fully addressed.

---

### Author Response · Authors · 2025-12-03

We thank the reviewers once again for their careful reading and constructive feedback. To conclude the discussion phase, we first highlight the key positive aspects of the work that reviewers emphasized:
- Theory shows the method is close to the best possible, and tests on internet traffic traces cut error by twenty to fifty percent versus standard sketches (Reviewer cfdo)
- Novel and timely problem formulation (Reviewer cfdo)
- First learning-augmented guarantees for time-decay streams (Reviewer cfdo)
- Captures privacy-driven data deletion (GDPR) and recency-weighted analytics, motivating practical relevance (Reviewer cfdo)
- Theoretically strong results (Reviewer cfdo)
- General reduction shows any ($\alpha$, $\beta$)-smooth function enjoys a black-box sliding-window lift while preserving approximation and randomised guarantees; rectangle and cascaded norms benefit for free (Reviewer cfdo)
- Experimental validation on real data: CAIDA traces (30 M IPs) show AMSA (augmented AMS) keeps relative error ≤1.2× across window sizes while AMS drifts up to 2.3× (Reviewer cfdo)
- Distribution-shift experiment on synthetic data demonstrates >2× error gap between SSA and SS when stream distribution changes, confirming robustness claim (Reviewer cfdo)
- Code released aiding reproducibility (Reviewer cfdo)
- Experiments on both real-world and synthetic datasets demonstrate the practical efficiency of the proposed approach (Reviewer szXb)
- The extension of learning-augmented frequency moment estimation to general time-decay settings is novel and appealing (Reviewer szXb)
- The paper provides space complexity bounds for several common estimation tasks...filling an existing theoretical gap (Reviewer szXb)
- The experiments convincingly show that the learning-augmented approach significantly improves estimation accuracy under the sliding-window model (Reviewer szXb)
- Moment estimation is an important problem sketching problem and I find it well-motivated in the learning augmented-framework. It is well-studied in several past works (Reviewer zdR7)
- The sliding window model is also natural since it captures the idea that we may often be more interested in statistics of the most recent data  (Reviewer zdR7)
- I like specifically about the paper, that they generalize the framework of Braverman and Ostrovsky [BOO7] to more general time-decay functions  (Reviewer zdR7)
- It is well-motivated to study the learning-augmented streaming under time-decay weighting (Reviewer oP5F)
- The paper proposes a unified reduction framework from streaming to time-decay models, and obtains near optimal algorithm for fundamental frequency estimation problems (Reviewer oP5F)

We then summarize the clarifications, additional experiments, and revisions we incorporated in the uploaded revised manuscript, with all changes marked in blue, which directly address the initial concerns raised:

- Expanded and rigorized key proofs (incl. rewritten Lemma 3.1), formalized the smooth-histogram sandwiching argument, and clarified how suffix compatibility ensures correctness across all instances.
- Improved algorithm exposition with clearer notation, corrected indexing, explicit return values, and an illustrative histogram figure.
- Added new experiments for \((k,p)\)-cascaded norms (3,3) and (5,5), expanding beyond the original \(\ell_2/\ell_3\) setting, along with CPU and peak-memory benchmarks.
- Clarified parameter files (seeds, repetitions, bucket sizes, sampling probabilities, prefix lengths) to improve reproducibility.
- Sharpened model assumptions, including the insertion-only update rule and frequency-vector initialization.
- Expanded explanation of decay models and how $(\eta,\nu)$ instantiate for polynomial vs. exponential decay.
- Clarified why comparison to prior learning-augmented sliding-window algorithms (e.g., Shahout et al.) is not meaningful.
- Added missing definitions (e.g., $x_A$) and standardized notation (e.g., $\mathrm{ALG}(k)$).
- Improved clarity in reviewer-flagged sections and incorporated additional citations.

In conclusion, we hope the revised manuscript more clearly conveys the contributions and insights of our work. Specifically, we provide the first learning-augmented guarantees for general time-decay streaming, introduce a unified framework that handles a variety of decay functions, and formalize suffix-compatible oracles to ensure correctness and efficiency across all algorithm instances. Our extensive experiments on both synthetic and real-world datasets show that the approach is robust to distribution shifts and consistently improves estimation accuracy compared to baseline methods. We believe these updates make the main ideas, theoretical innovations, and practical relevance of our work more transparent and accessible, while opening avenues for future research in learning-augmented streaming algorithms.

---

### Meta-Review · Area_Chair_irwC · 2025-12-19

**Summary:**

The paper considers learning augmented algorithms for moment estimation in time decay models. It builds on a recent interesting line of work on learning augmented algorithms, which take advice from an oracle to improve on worst-case guarantees.

The advice in the paper comes from (suffix-compatible) heavy-hitter oracles. The overall approach maintains several copies of the steaming algorithm as in Braverman & Ostrovsky (2007), and then applies the learning augmented framework of Jiang et al. (2020). The approach comes with theoretical guarantees, and is also validated on experiments.

Overall, this paper is borderline, but can be accepted if there is room. The problem is well-motivated, the contributions are solid, and with the revisions that the authors have done the writing is reasonable too. See some additional discussion of the concerns below.

**Reviewer Concerns:**

The major concerns were from reviewers zdR7 and oP5F.

Reviewer zdR7 has two major concerns, regarding the writing and lack of rigor in proofs, and regarding technical novelty. The point regarding technical novelty more or less stands and is not addressed, but perhaps we don't need to judge the paper too harshly for this. The paper does identify a useful oracle (the suffix compatible heavy hitter oracle) which enables the guarantees, and the results are good. The concern regarding writing is unfortunate on part of the authors. They did not fix the concerns that the reviewer had in Neurips, and kept the proofs very sloppy. They have since fixed the writing and it reads better now, but in some sense they really should have submitted a better version so that it could be reviewed properly. Reviewer zdR7's concerns are why I think the paper is still only borderline at best.

Reviewer oP5F had concerns about how the oracle is implemented in theory and practice, and these appear to be addressed.

**Reviewer Scores:**

Reviewer zdR7 would probably keep their score, Reviewer oP5F may raise. Rest may keep their score too.

---

### Decision · Program_Chairs · 2026-01-26

Accept (Poster)